# Statistical analysis of dynamic behavior of continental shelf wave motions in the northern South China Sea

**Junyi Li**[1,2,3]**, Tao He**[1,3]**, Quanan Zheng**[1,4]**, Ying Xu**[3]**, and Lingling Xie**[1]

[1]Laboratory of Coastal Ocean Variation and Disaster Prediction, College of Ocean and Meteorology, Guangdong Ocean University, Zhanjiang 524088, China
[2]Key Laboratory of Climate, Sources and Environments in Continent Shelf Sea and Deep Ocean, Zhanjiang 524088, China
[3]Key Laboratory of Space Ocean Remote Sensing and Application, MNR, Beijing, 100081, China
[4]Department of Atmospheric and Oceanic Science, University of Maryland, College Park, MD 20742, USA

**Correspondence:** Lingling Xie (xiell@gdou.edu.cn)

**Abstract.** This study aims to analyze statistical behavior of the continental shelf wave motions, including continental shelf waves (CSWs) and arrested topographic waves (ATWs), in the northern South China Sea. The baseline consists of tide-gauge data from stations Kanmen, Xiamen, Shanwei, Hong Kong CE1, and Zhapo as well as along-track sea level anomaly (SLA) data derived from multiple satellite altimeters from 1993 to 2020. The subtidal signals propagating along the coast with periods shorter than 40 d and phase speeds of about $10 \, \text{m s}^{-1}$ are interpreted as CSWs. The cross-shelf structure of along-track SLAs indicates that Mode 1 of CSWs is the predominant component trapped in the area shallower than about 200 m. The amplitudes of CSWs reach a maximum of 0.6 m during July–September and a minimum of 0.2 m during April–June. The inter-seasonal and seasonal signals represent ATWs. The amplitudes of ATWs reach 0.10 m during October–December, twice that during July–September. These observations can be well interpreted in the framework of linear wave theory. The cross-shelf structures of CSWs and ATWs derived from along-track SLAs illustrate that the methods are suitable for observing dynamic behavior of the CSWs.

## 1 Introduction

The continental shelf wave (CSW) is a type of topographic Rossby wave (TRW) trapped on the continental shelf with amplitudes ranging from several tens of centimeters to more than 1 m (Aydın and Beşiktepe, 2022; Clarke and Brink, 1985; Heaps et al., 1988; Morey et al., 2006; Mysak, 1980; Robinson, 1964; Zheng et al., 2015). A CSW is a sub-inertial motion with a wavelength much greater than the depth (Li et al., 2015; Schulz et al., 2011). It propagates along the shelf with the coast on its right (left) in the Northern (Southern) Hemisphere (Clarke, 1977). During the impact of a typhoon, excessive flooding in the coastal zone could be induced by a propagating CSW that is added to the locally wind-generated surge (Dukhovskoy and Morey, 2011; Han et al., 2012). Therefore, the CSW is particularly important for coastal sea level variations.

A CSW is generally generated by large-scale weather systems moving across or along the shelf (Thiebaut and Vennell, 2010). CSW events reported by previous investigators lasted from 2 d to 2 weeks (Chen and Su, 1987; Li et al., 2015, 2021; Zheng et al., 2015). The phase speed of CSWs depends on the bottom topography, ranging from 5 to $20 \, \text{m s}^{-1}$ (Li et al., 2015, 2016; Shen et al., 2021). CSWs could be taken as barotropic motions in an unstratified coastal zone, while in a stratified ocean, the response should be classified as coastal trapped waves. Overall, they result from conserving potential vorticity over the shelf (Chen et al., 2022; Quan et al., 2021; Wang and Mooers, 1976).

The sea level variations in the South China Sea (SCS) are depicted well by these previous studies as the continental shelf occupies about half its area: Ding et al. (2012), Li et al. (2023b), Shen et al. (2021), Zhao et al. (2017) and Zhou et al. (2023). However, two issues should be improved.

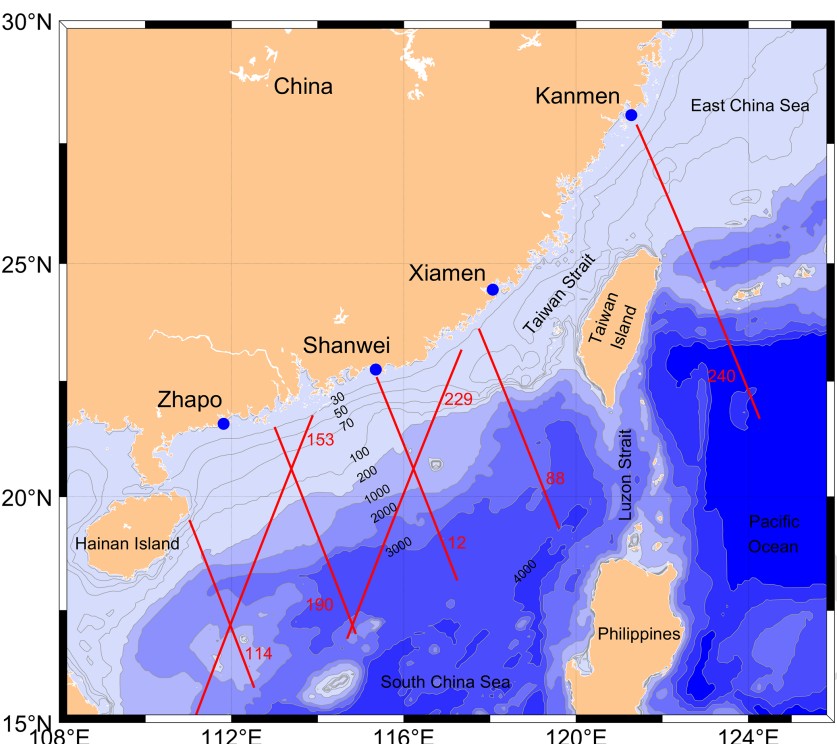

**Figure 1.** Study area. Blue dots represent locations of tide-gauge stations Kanmen, Xiamen, Shanwei and Zhapo. Red lines represent segments of ground tracks 12, 88, 114, 153, 190, 229 and 240 for altimeter satellites over the continental shelf. Tracks 12, 88,114 and 190 are almost perpendicular to the coastline. Isobaths are in meters.

The first is that the primary data are usually obtained from the tide-gauge stations along the coastline. The data have high accuracy but represent the sea level at the coasts only. Thus, satellite-altimeter-observed sea level variations are often used to fill the data gaps in between the tide-gauge stations and on the continental shelf. The second issue is that the repeat period of the satellite altimeter is 9.9 d, which makes it challenging to investigate sea level variations with periods shorter than 10 d. Previous studies used the along-track sea level anomaly (SLA) from satellite altimeters to describe CSWs (Chen et al., 2014; Li et al., 2016). However, the satellite altimeter with sparse tracks (as shown in Fig. 1) could only capture the cross structure of CSWs with one or two snapshots of one CSW.

Against the background of global warming, sea level variation with very low frequency has been investigated previously. Ho et al. (2000) found seasonal sea level variability in the SCS using data from a satellite altimeter. Kajikawa and Yasunari (2005) investigated the interannual variability in the intra-seasonal variation over the SCS. Fang et al. (2006) analyzed low sea level along the eastern boundary of the SCS (Fang et al., 2006). Zhuang et al. (2010) found strong intra-seasonal variability in the northern SCS. In addition, sea level variations are influenced by thermodynamic processes, e.g., eddies and thermal change in the upper layer of the SCS (Cheng and Qi, 2007; Xie et al., 2018; Zheng et al., 2014).

Meanwhile, the upper-layer thermal changes significantly influence sea level variations (Cheng and Qi, 2007). Using sea surface height (SSH) data from satellite altimeters, Xu et al. (2016) found that sea level variations in the coastal area of the SCS are still strongly influenced by the coastal current system in summer and winter. Seasonal circulation is mainly driven by monsoon wind stress (Gan et al., 2006). Lin et al. (2021) applied the arrested topographic wave (ATW) model to the coastal mean dynamic topography along the East China Sea (ECS) and the SCS. The mean circulation in a coastal zone of variable depth may be modeled by linear equations (Wu, 2021). The result suggests that the mean dynamic topography is a counterbalance of contributions from the along-shelf wind and bottom friction and is predicted well by the ATW model. Therefore, monsoon winds are a control factor for sea level variation.

This study aims to investigate the cross-shelf structures of sea level over the continental shelf. As the repeat period of the satellite altimeters, 9.9 d, is comparable with that of CSWs in the northern SCS, the statistical characteristics of the along-track SLA are applied to show the cross-shelf structure of CSWs using a long-term dataset from 1993 to 2020. To figure out the cross-shelf structures of ATWs is another goal.

The rest of the paper is organized as follows: Sect. 2 describes the observed data. Section 3 presents the theory of

CSWs and ATWs. Section 4 presents the characteristics of the signals derived from the tide-gauge data and along-track SLA. Section 5 discusses the CSWs detected from tide-gauge data and the cross-shelf structure of sea level over the conti-
5 nental shelf. Section 6 gives summaries.

## 2  Data

### 2.1  Along-track sea level anomaly data

Satellite altimeter along-track SLA data are produced and distributed by the Archiving, Validation, and Interpretation
of Satellite Oceanographic Data (AVISO), Centre National d'Etudes Spatiales (CNES) of France. The data from 1993 to 2020 are derived from TOPEX/Poseidon, Jason-1, Jason-2 and Jason-3 measurements. The satellite repeat period is 9.9 d, and the temporal resolution of the along-track data is
1 Hz. The along-track SLA is calculated by subtracting the 20-year mean from the SSH measured by the satellite altimeters. The along-track SLA is low pass filtered using a seven-point moving average. The ground tracks in the study area, 12, 88, 114, 153, 229 and 240, are shown in Fig. 1.
Satellite altimetry provides a unique sea level dataset for coastal sea level research. A few recent studies have stressed the importance of small-scale coastal processes for coastal sea level variance (Cazenave and Moreira, 2022; Vignudelli et al., 2019). The along-track SLA has been successfully
validated and applied to the coastal zone by Birol et al. (2021). The studies mentioned in this paragraph present the availability of along-track SLA in the coastal zones.

### 2.2  Sea level anomaly from tide gauges

The tide-gauge data at stations Kanmen, Xiamen, Shanwei,
Hong Kong and Zhapo (as shown in Fig. 1) are obtained from the Global Sea Level Observing System (GLOSS). The data cover a period from 1993 to 1997, with a temporal resolution of 1 h. The de-tided sea level anomaly (DSLA) is calculated by removing tidal signals using a MATLAB toolbox
(Pawlowicz et al., 2002).
  The monthly sea level means at stations Xiamen, Shanwei and Zhapo are obtained from the Permanent Service for Mean Sea Level (PSMSL). Monthly mean data cover the periods of 1993–2003, 1993–1994 and 1993–2020 for the sta-
40 tions Xiamen, Shanwei and Zhapo, respectively.

### 2.3  Sea surface wind stress

Monthly sea surface wind stress is derived from the Copernicus Marine Environment Monitoring Service (CMEMS). The dataset covers a period from 1993 to 2020 with a spa-
45 tial resolution of $0.25° \times 0.25°$. Sea surface wind stress data on the satellite altimeter ground tracks are decomposed into the cross-shelf and along-shelf components. The cross-shelf component is positive seaward and parallel to the satellite al-

**Table 1.** The width of the continental shelf ($l$) and the depths of the shelf break ($H_1$) and deep basin ($H_2$) along the satellite ground tracks.

| Track number | $l$ (km) | $H_1$ (m) | $H_2$ (m) |
|---|---|---|---|
| 12 | 200 | −300 | −3800 |
| 88 | 178 | −200 | −2500 |
| 114 | 123 | −200 | −1800 |
| 153 | 259 | −110 | −1600 |
| 190 | 247 | −255 | −3500 |
| 229 | 244 | −225 | −3500 |
| 240 | 273 | −250 | −5000 |

timeter ground tracks (12, 88, 114 and 190). The along-shelf component is positive northward and perpendicular to these 50 satellite altimeter ground tracks.

### 2.4  Topographic profile

The behavior of CSWs is determined by the topography of the continental shelf and slope, which has been well documented. The topographic profiles along the satellite altimeter 55 ground tracks are extracted from a dataset of ETOPO-2. This study uses one-dimensional linear piecewise functions to fit the topographic profiles along the satellite altimeter ground tracks. The width of the continental shelf, depths of shelf break and deep basin along the tracks are listed in Table 1. 60 The continental shelf break is extracted as the location of maximum change in the gradient of the continental slope.

## 3  Theory

### 3.1  Momentum equation for CSWs

The linearized shallow-water equations governing a 65 barotropic ocean on a rotating earth are

$$\frac{\partial u}{\partial t} - f v = -g \frac{\partial \eta}{\partial x} + \frac{\tau_s^x - \tau_b^x}{\rho H}, \tag{1a}$$

$$\frac{\partial v}{\partial t} + f u = -g \frac{\partial \eta}{\partial y} + \frac{\tau_s^y - \tau_b^y}{\rho H}, \tag{1b}$$

$$\frac{\partial \eta}{\partial t} + \frac{\partial (uH)}{\partial x} + \frac{\partial (vH)}{\partial y} = 0, \tag{1c}$$

where cross-shelf and along-shelf velocities ($u$, $v$) are depth- 70 averaged in cross-shelf and along-shelf coordinates ($x$, $y$). $\eta$ is the sea surface height. The Coriolis parameter is $f$. The bathymetry $H = H(x)$ is assumed to be a function of the cross-shelf variable, $x$ only. $\tau_s$ and $\tau_b$ are the surface and bottom stresses. $g$ and $\rho$ are the gravitational acceleration and 75 the water density.
  The scales of the along-shelf length of CSW ($L = 2\pi/k \approx 2 \times 10^3$ km) and the cross-shelf length ($l \approx 200$ km) are subject to the long-wave assumption ($l/L \ll 1$). Under the long-

wave assumption, scaling is applied to Eq. (1a and b). Then, $\partial u / \partial t \ll f v$. We neglect the $\partial u / \partial t$ term, and Eq. (1a and b) become (Li et al., 2016; Schulz et al., 2011):

$$v = \frac{g}{f} \frac{\partial \eta}{\partial x} - \frac{\tau_s^x - \tau_b^x}{f \rho H}, \tag{2a}$$

$$u = -\frac{g}{f} \frac{\partial \eta}{\partial y} - \frac{\partial}{\partial t} \left( \frac{g}{f^2} \frac{\partial \eta}{\partial x} - \frac{\tau_s^x - \tau_b^x}{f^2 \rho H} \right) + \frac{\tau_s^y - \tau_b^y}{f \rho H}. \tag{2b}$$

Substituting Eq. (2a and b) into Eq. (1c), we obtain the equation governing SSH of CSWs:

$$\frac{\partial}{\partial t} \left( \eta - \frac{gH}{f^2} \frac{\partial^2 \eta}{\partial x^2} - \frac{g}{f^2} \frac{\partial \eta}{\partial x} \frac{dH}{dx} \right)$$
$$- \frac{g}{f} \frac{\partial \eta}{\partial y} \frac{dH}{dx} + \frac{\partial}{\partial x} \left( \frac{\tau_s^y - \tau_b^y}{f \rho} \right) + \frac{\partial^2}{\partial t \partial x} \left( \frac{\tau_s^x - \tau_b^x}{f^2 \rho} \right)$$
$$- H \frac{\partial}{\partial y} \left( \frac{\tau_s^x - \tau_b^x}{f \rho H} \right) = 0. \tag{3}$$

Assume that CSWs are forced by along-shelf wind stress, i.e., $(\tau_s^x, \tau_s^y) = (0, \tau_s^y)$. The bottom friction is neglected to simplify the calculation in Eq. (3). Therefore, the equation governing the SSH of CSWs becomes

$$\frac{\partial}{\partial t} \left( \eta - \frac{gH}{f^2} \frac{\partial^2 \eta}{\partial x^2} - \frac{g}{f^2} \frac{\partial \eta}{\partial x} \frac{dH}{dx} \right)$$
$$- \frac{g}{f} \frac{\partial \eta}{\partial y} \frac{dH}{dx} + \frac{\partial}{\partial x} \left( \frac{\tau_s^y}{f \rho} \right) = 0. \tag{4}$$

The change in the SSH of CSWs is balanced by the variation in along-shelf wind stress in a cross-shelf direction. Assume a periodic along-shelf wind stress, $\tau_s^y = \tau_0 \exp[i(\alpha y + \omega t)]$ (where $\alpha$ is the wavenumber of wind stress and $\tau_0 = $ constant). Equation (4) becomes

$$\frac{\partial}{\partial t} \left( \eta - \frac{gH}{f^2} \frac{\partial^2 \eta}{\partial x^2} - \frac{g}{f^2} \frac{\partial \eta}{\partial x} \frac{dH}{dx} \right) - \frac{g}{f} \frac{\partial \eta}{\partial y} \frac{dH}{dx} = 0, \tag{5}$$

which means that the change in SSH is independent of along-shelf wind stress.

The assumption of a wave solution,

$$\eta = \phi(x) \exp[i(ky + \omega t)], \tag{6}$$

yields an equation for $\phi(x)$:

$$H \frac{d^2 \phi}{dx^2} + \frac{dH}{dx} \frac{d\phi}{dx} + \left( \frac{fk}{\omega} \frac{dH}{dx} - \frac{f^2}{g} \right) \phi = 0. \tag{7}$$

### 3.1.1 Over the shelf

For $\leq x \leq l$, $H = H_1 x / l$, the equation for $\phi(x)$ is

$$x \frac{d^2 \phi}{dx^2} + \frac{d\phi}{dx} + \left( \frac{fk}{\omega} - \frac{f^2 l}{gH_1} \right) \phi = 0, \tag{8}$$

which is subject to the following boundary conditions: $\phi(0) = a$ and $\phi(l) = A$. $a$ and $A$ should be arbitrary. We could take $a$ as the amplitude of fluctuation in sea level from the tidal gauge station.

The solution to Eq. (8) is expressed as the sum of the first and second kinds of Bessel functions (Robinson, 1964; Schulz et al., 2011):

$$\phi(x) = a J_0 \left( 2 \left( \frac{fk}{\omega} - \frac{f^2 l}{gH_1} \right)^{\frac{1}{2}} x^{\frac{1}{2}} \right)$$
$$+ b Y_0 \left( 2 \left( \frac{fk}{\omega} - \frac{f^2 l}{gH_1} \right)^{\frac{1}{2}} x^{\frac{1}{2}} \right), \tag{9}$$

where $a$ is arbitrary constant. As the solution for $\phi(x)$ is finite, therefore

$$\phi(x) = a J_0 \left( 2 \left( \frac{fk}{\omega} - \frac{f^2 l}{gH_1} \right)^{\frac{1}{2}} x^{\frac{1}{2}} \right). \tag{10}$$

### 3.1.2 In the deep basin

For $l < x$, $H = H_2$, the equation for $\phi(x)$ is

$$\frac{d^2 \phi}{dx^2} - \frac{f^2}{gH_2} \phi = 0, \tag{11}$$

which is subject to the following boundary conditions: $\phi(x \to l) = A$ and $\phi(\infty) = 0$.

The solution is

$$\phi(x) = A \exp \left( -\frac{fl}{\sqrt{gH_2}} \left( \frac{x}{l} - 1 \right) \right)$$
$$+ B \exp \left( \frac{fl}{\sqrt{gH_2}} \left( \frac{x}{l} - 1 \right) \right), \tag{12}$$

as the solution for $\phi(x)$ is also finite in the deep basin; i.e.,

$$\phi(x) = A \exp \left( -\frac{fl}{\sqrt{gH_2}} \left( \frac{x}{l} - 1 \right) \right), \tag{13}$$

where $A$ is an arbitrary constant.

### 3.1.3 Dispersion relation

As the fluid is continuous at the edge of continental shelf, therefore

$$H_1 \cdot u|_{x \to l_-} = H_2 \cdot u|_{x \to l_+}, \tag{14a}$$

$$a J_0 \left( 2 \left( \frac{fkl}{\omega} - \frac{f^2 l^2}{gH_1} \right)^{\frac{1}{2}} \right) = A. \tag{14b}$$

From Eq. (14a), we have

$$H_1 \cdot a \left( k J_0 + \frac{\omega}{f} \left( \frac{fk}{\omega l} - \frac{f^2}{gH_1} \right)^{\frac{1}{2}} J_0' \right)$$
$$= H_2 \cdot A \left( k - \frac{\omega}{\sqrt{gH_2}} \right), \tag{15}$$

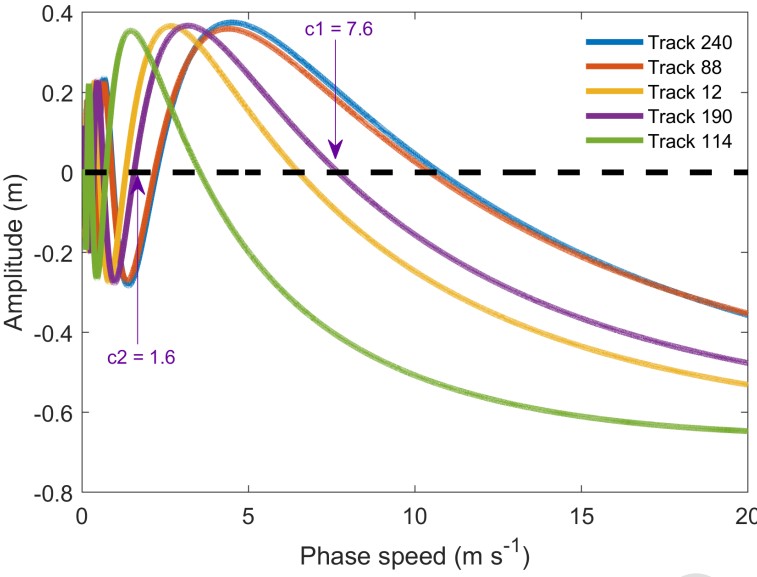

**Figure 2.** Solution of phase speed for CSWs from Eq. (17). The zero-crossing values represent the phase speed of CSWs for different modes. Colorful curves are the amplitude of the zero order of the first kind of Bessel function in different modes for the idealized depth profile of tracks 12, 88, 114, 190 and 240. Arrows indicate the phase speed of Mode 1 ($7.6\,\mathrm{m\,s^{-1}}$) and Mode 2 ($1.6\,\mathrm{m\,s^{-1}}$) CSWs along track 190.

where $J_0'$ is the derivation of the zero order of the first kind of Bessel function:

$$J_0'(x) = -\frac{x}{2}(J_0(x) + J_2(x)), \qquad (16)$$

where $J_2$ is the second-order Bessel function of the first kind.

Substituting Eq. (16) into Eq. (15) yields a dispersion relation for CSWs:

$$\left(\frac{cfl}{gH_2} - 1 + \frac{c}{(gH_2)^{\frac{1}{2}}}\right) J_0 - \left(\frac{H_1}{H_2} - \frac{cfl}{gH_2}\right) J_2 = 0, \qquad (17)$$

where $c(= \frac{\omega}{k})$ is the phase speed of CSWs. $J_0$ and $J_2$ are known and balanced by the characteristics of CSWs and topography. We solve Eq. (17) using the zero-finding function in MATLAB. The solution of the phase speed for CSWs is shown as Fig. 2. The zero-crossing points for each curve present the phase speed of CSWs. The first zero-crossing point on the right hand of each curve indicates the phase speed of Mode 1 CSWs, e.g., $c = 7.6\,\mathrm{m\,s^{-1}}$ for track 190.

### 3.1.4 Cross-shelf structure

With Eqs. (10), (13) and (14b), the wave solution of SSH, i.e., Eq. (3), is

$$\eta = \begin{cases} \sum_{i=1}^{\infty} a_i \cdot J_0\left(2\left(\frac{fk_i}{\omega_i} - \frac{f^2 l}{gH_1}\right)^{\frac{1}{2}} x^{\frac{1}{2}}\right) \\ \times \exp[i(k_i y + \omega_i t)] & x \le l \\ \sum_{i=1}^{\infty} a_i \cdot J_0\left(2\left(\frac{fk_i l}{\omega_i} - \frac{f^2 l^2}{gH_1}\right)^{\frac{1}{2}}\right) \\ \times \exp\left(-\frac{fl}{\sqrt{gH_2}}\left(\frac{x}{l} - 1\right)\right) \exp[i(k_i y + \omega_i t)] & x > l, \end{cases}$$

$$(18)$$

where $\exp[i(ky + \omega t)]$ is the waveform propagating along the shelf. $a_i \cdot J_0$ is the cross-shelf structure of the waveform for Mode $i$ over the shelf.

In the cross-shelf direction, the SSH becomes

$$\eta = \begin{cases} \sum_{i=1}^{\infty} \eta_{0i}(t) \cdot J_0\left(2\left(\frac{fk_i}{\omega_i} - \frac{f^2 l}{gH_1}\right)^{\frac{1}{2}} x^{\frac{1}{2}}\right) \\ \qquad\qquad x \le l \\ \sum_{i=1}^{\infty} \eta_{0i}(t) \cdot J_0\left(2\left(\frac{fk_i l}{\omega_i} - \frac{f^2 l^2}{gH_1}\right)^{\frac{1}{2}}\right) \\ \times \exp\left(-\frac{fl}{\sqrt{gH_2}}\left(\frac{x}{l} - 1\right)\right) \qquad x > l, \end{cases}$$

$$(19)$$

where $\eta_{0i}(t)(= a_i \exp[i(k_i y + \omega_i t)])$ is time series of SSH at the coastline for Mode $i$.

Figure 3 shows the SSH structure over the continental shelf. The SSH Mode 1 looks like a bell mouth. For modes 2 and 3, nodes and antinodes appear on the shelf. Nodes for Mode 2 appear at 50 km off the coast over the shelf. Nodes for Mode 3 appear at 30 and 90 km off the coast.

### 3.2 Steady situation

The governing equations in the steady situation for Eqs. (1a-c) are

$$-fv = -g\frac{\partial \eta}{\partial x}, \qquad (20a)$$

$$fu = -g\frac{\partial \eta}{\partial y} + \frac{\tau_s^y - \tau_b^y}{\rho H}, \qquad (20b)$$

$$\frac{\partial(uH)}{\partial x} + \frac{\partial(vH)}{\partial y} = 0. \qquad (20c)$$

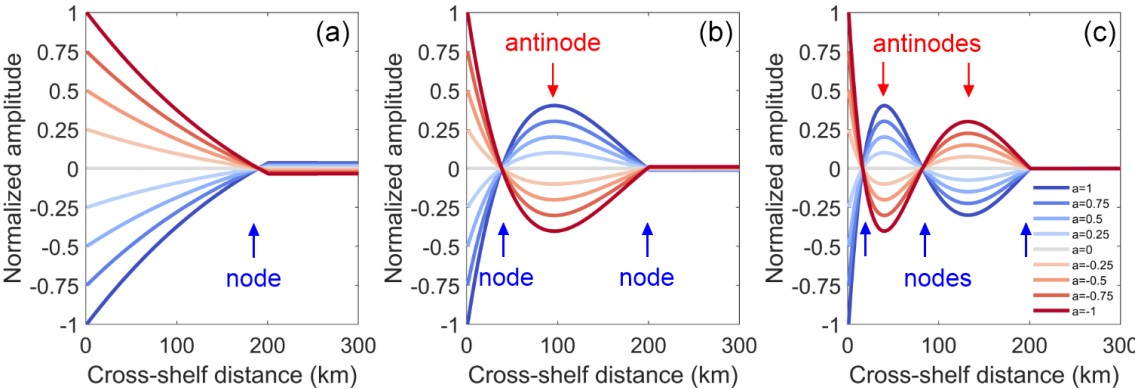

**Figure 3. (a–c)** Cross-shelf structure of normalized amplitudes of the first three CSW modes for the idealized depth profile of track 12. Gradient color curves represent amplitude structure over time for CSWs in the cross-shelf direction.

A linear drag is used for the bottom friction; i.e., $\boldsymbol{\tau_b} = \rho\lambda\boldsymbol{u}$ (Hsueh and Pang, 1989; Lin and Yang, 2011). The solution for SSH based on Eqs. (20a–c) gives (Csanady, 1978)

$$\eta = B\tau_0 e^{-\frac{x}{L}} \sin\left(\alpha y + \frac{x}{L} - \frac{\pi}{4}\right), \qquad (21)$$

where $L = \sqrt{2\lambda l}/\alpha f H_1$ is the scale width of trapped sea level in the cross-shelf direction and $B$ is an arbitrary constant. A drag coefficient, $\lambda = O(5 \times 10^{-4})\,\mathrm{m\,s^{-1}}$, is used for the linear bottom friction (Chapman, 1987; Lin and Yang, 2011). The typical magnitude for winter wind stress over the northern continental shelf of the SCS is $O\,(0.1)\,\mathrm{N\,m^{-2}}$ (Lin and Yang, 2011; Lin et al., 2011). $f = 5 \times 10^{-5}\,\mathrm{s^{-1}}$, and $\alpha = 2\pi/4000\,\mathrm{km}$, evaluated by Lin et al. (2021).

The normalized SSH in the cross-shelf direction of ATW for tracks 12, 88, 114 and 190 is shown in Fig. 4. One can see that the trapped sea level in the cross-shelf direction decays quickly from 1 at the coastline to $\sim 0.2$ at the edge of the continental shelf ($\sim 200\,\mathrm{km}$) and 0.1 at a distance of 300 km. The ATW amplitude decays offshore, and $L = \sim 100\,\mathrm{km}$ in the study area, which is much less than the local Rossby radius of deformation ($\sim 600\,\mathrm{km}$). Under different wind stresses, the amplitude of ATW evolves similarly to that in Fig. 3a. As track 240 is not perpendicular to the coastline, it is beyond the scope of this paper.

## 4 Signals in sea level anomaly

### 4.1 Tide-gauge data

Figure 5 shows the wavelet transform (WT) of the SLA at station Xiamen from 1993 to 1997. One can see abundant signals with periods from several days to 1 year in Fig. 5a and b. A dozen signals with periods from several days to about 1 month could be seen from the WT of SLA every year. The duration of these signals, with a period of several days, is about 10 d (e.g., in Fig. 5d). Moreover, the signals with a period of about 1 month are sustained for about 3 months

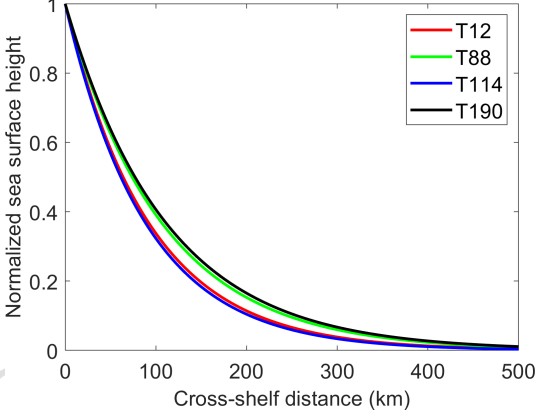

**Figure 4.** Normalized SSH in a cross-shelf direction for tracks 12, 88, 114 and 190.

(Fig. 5c and d). Even if the significance level against red noise is larger than 5 %, the power is universal and continuous in the period bands of 10–60 d. That is, these signals are not so significant compared with the signals with large amplitude. The wavelet analysis also exhibits a significant inter-seasonal and seasonal variation in the SLA. The characteristics of the signals at stations Kanmen, Shanwei and Zhapo are almost the same (not shown here). Therefore, the variation in the SLA along the northern SCS coast is universal.

Figure 5c and d show the cross wavelet transform (XWT) of the SLA between stations Kanmen and Xiamen. One can see that the period band and the occurrence time of the significant cross wavelet power are consistent with those in Fig. 5a. The signals with periods shorter and longer than 40 d show remarkably different characteristics. In the period band shorter than 40 d, one can see the arrows generally point down and to the right. The quasi-uniform phase lag indicates the fixed time delay of the signal between two tidal gauge stations. However, the direction of arrows is disorderly for

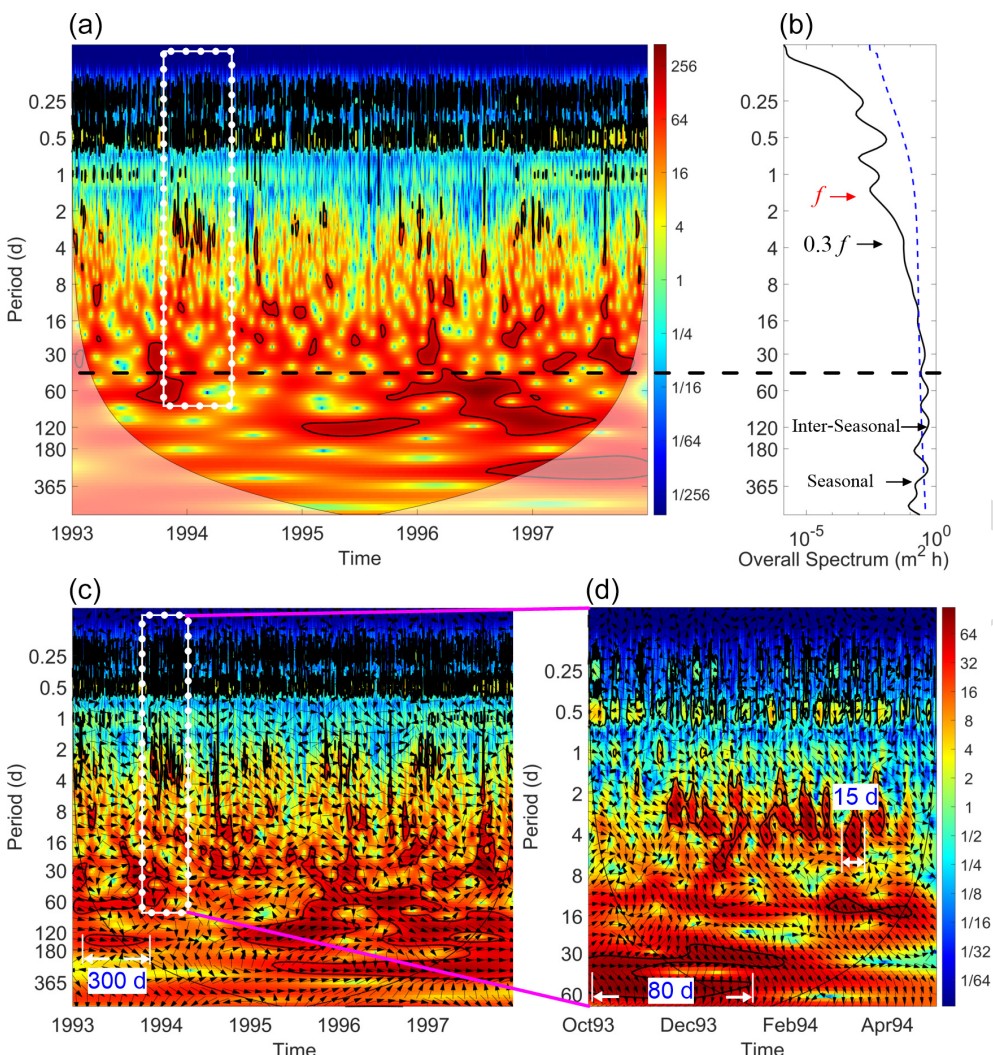

**Figure 5.** Temporal variability in the SLA at the coastline. **(a)** The WT of the SLA at station Xiamen. **(b)** Overall spectrum of the SLA at station Xiamen. **(c)** The XWT of the SLA between stations Kanmen and Xiamen. **(d)** The XWT of the SLA between stations Kanmen and Xiamen from October 1993 to April 1994. White line and arrows are auxiliary lines indicating duration. The thick black contour in **(a)**, **(c)** and **(d)** is a 5 % significance level against red noise, and the cone of influence (COI) is shown as a thin line. Color codes of power spectra normalized by variance are in arbitrary units. The white dotted rectangles in **(a, c)** show the temporal domain of **(d)**. The blue dashed curve in **(b)** is a 5 % significance level. Arrows in **(b)** indicate the characteristic frequencies of inertial oscillation and CSWs. The black dashed line in **(a, b)** represents the signal period boundary (40 d). The arrows in **(c, d)** show the relative phase relationship between the SLA at Kanmen and Xiamen with in-phase (anti-phase, leading and lagging) pointing right (left, down and up). In **(d)**, arrows point down and to the right (about $\pi/3$), indicating that the SLA in Kanmen is leading that in Xiamen.

the period band longer than 40 d, which indicates that there is no evidence of propagation.

In the period band shorter than 40 d, the signals at station Xiamen lag that at station Kanmen by about 15 h (time lag = phase difference $/2\pi \times$ period of signal). The propagation phase speed of the sea level signal could be calculated by the lag time of sea level propagation between stations Kanmen and Xiamen. The result is about 9 m s$^{-1}$, which is very close to that reported by a number of recent studies (Ding et al., 2012; Li et al., 2015, 2016; Zhao et al., 2017).

In the period band longer than 40 d, the phase of signals between Kanmen and Xiamen is a little complicated. The signal phase with the period of 360 d from 1995 to 1997 is almost 0. However, the seasonal signals at station Xiamen lag (lead) that at station Kanmen in winter (summer) by about $\pi/4 - \pi/2$, implying that the signal propagates very slowly along the coast. Csanady (1978) concluded that such signals are a kind of ATW.

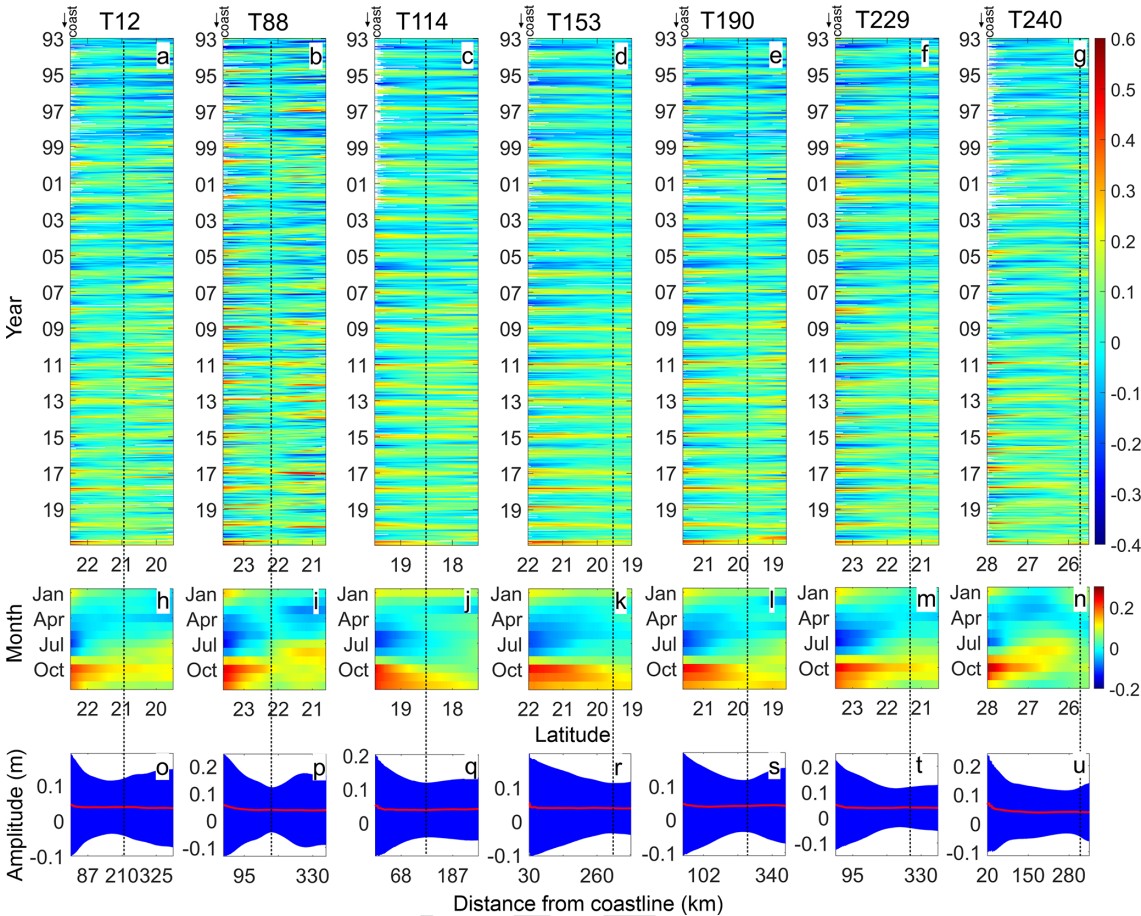

**Figure 6. (a–g)** Latitudinal distribution of the along-track SLA over the northern continental shelf of the SCS from 1993 to 2020 for tracks 12, 88, 114, 153, 190, 229 and 240. **(h–n)** Climatological monthly mean of the along-track SLAs for tracks 12, 88, 114, 153, 190, 229 and 240. **(o–u)** Mean (red curves) and standard deviation (blue shading) of the along-track SLA from 1993 to 2020 for tracks 12, 88, 114, 153, 190, 229 and 240. Track numbers are shown at the tops of panels. Vertical dashed lines represent shelf break positions along the tracks, i.e., $H_1$, listed in Table 1. The coastline position is marked by an arrow at the top left of each panel.

## 4.2 Along-track SLA

Figure 6 shows the latitudinal distribution of the along-track SLA over the northern continental shelf of the SCS from 1993 to 2020. One can see the clear annual signals near the coast in Fig. 6a–g. Signals with periods shorter than 1 year could also be seen. As shown in Fig. 6h–n, a clear seasonal cycle is discernable from the climatological monthly mean of the along-track SLA. One can see that lower (higher) sea levels over the shelf exist from March to August (September to February). The trough ($\sim -0.2$ m) and peak ($\sim 0.24$ m) of sea level near the coast occur in July and October, respectively, while in track 240, the climatological monthly mean of the along-track SLA on the shelf is smaller than that in track 88 especially in July. The 28-year mean value of the along-track SLA is about 0.04 m, as shown in Fig. 6o–u.

Moreover, the standard deviations (SDs) of the along-track SLA from 1993 to 2020 (blue shading in Fig. 6o–u) show a bell-mouth-like structure over the shelf. The amplitudes of the along-track SLA reach their maxima near the coast. The minimum variance exists near the shelf edge. The cross-shelf structure of the SLA indicates that sea level signals depend on the shelf depth.

## 5   Discussion

### 5.1   Propagation of CSWs

Figure 7 compares the data derived from the XWT of the SLA with the dispersion relation of CSW. We overlay the dispersion relation curves of CSWs with the results (phase speed, $c$, and period, $T$; $\lambda = c \cdot T$) from the tide-gauge data analysis in Sect. 4.1. One can see that the data points derived from station pairs are distributed near the dispersion relation of CSWs, implying the signals with the periods shorter than 40 d are the CSWs propagating along the shelf.

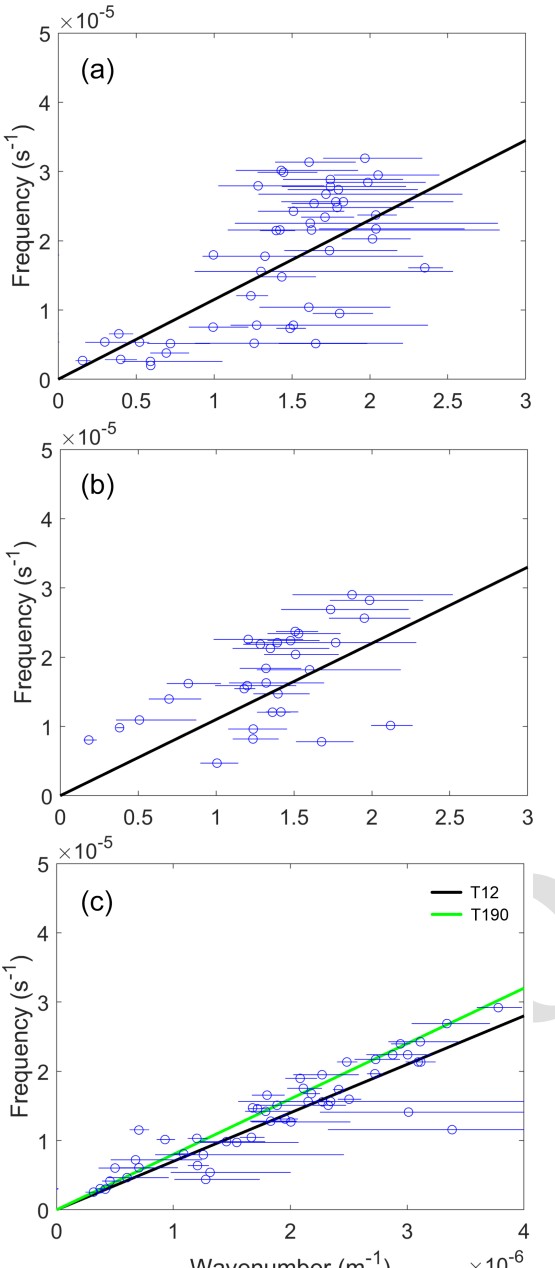

**Figure 7.** Dispersion relation of the lowest mode of CSWs between **(a)** Kanmen and Xiamen, **(b)** Xiamen and Shanwei, and **(c)** Shanwei and Zhapo. The data points are calculated from the XWT of the SLA. The curves are the theoretical dispersion relation for the mean depth profiles listed in Table 1. Black and green curves in **(c)** represent the dispersion relation (from Fig. 2) for the topographic profiles along tracks 12 and 190.

Figure 7a and b deal with the signals between Kanmen and Shanwei. We use the bathymetric profile near track 240 and 88 of the altimeter satellite to calculate the dispersion relation curve. However, since the topography between Kanmen and Shanwei changes dramatically, the data points lie dispersedly

around the curve. The wavenumbers of CSWs range from 0.1 to $2.4 \times 10^{-5}$ m$^{-1}$ between stations Kanmen and Shanwei. The CSWs propagate along the coast with a phase speed of 10 m s$^{-1}$.

Figure 7c shows the signals between Shanwei and Zhapo. The data points and the theoretical curves agree quite well. Compared with the topography between Kanmen and Shanwei, the topography changes slightly between stations Shanwei and Zhapo. Thus, we conclude that the signals in tide-gauge data with the periods shorter than 40 d are the CSWs propagating along the shelf. The phase speed of CSWs is about 8 m s$^{-1}$ in the study area, which is close to that of previous studies (Li et al., 2015, 2016; Shen et al., 2021).

Previous studies show that the periods of CSWs range from 2 d to 2 weeks in the SCS (Chen and Su, 1987; Li, 1993). The period of CSWs upstream, i.e., in the ECS, is often detected as several days (Ding et al., 2012, 2018; Hsueh and Pang, 1989; Huang et al., 2015; Yin et al., 2014). Hsueh and Romea (1983) found the sea level fluctuations with a period of more than 13 d along the west Korean coast. Worldwide, these low-frequency CSWs are common along the coast of Chile and the east coast of the Indian Ocean (Castro and Lee, 1995; Hormazabal et al., 2001; Marshall and Hendon, 2013; Vialard et al., 2009), where the width of the continental shelf is narrow. In this study, we define the CSW with a maximum period of about 30–40 d. The main reason for the difference is the data length we used. The long-time series of the DSLA helps analyze the abnormal low-frequency CSWs in the SCS.

In addition, we should take care to note that CSWs in Fig. 7 are mixed with wind-forced CSWs. It is difficult to separate the effect of wind-forced and freely propagating CSWs clearly. Hsueh and Romea (1983) found that there is a clear coupling between surface winds and coastal sea level in the northeast China Sea. Li et al. (2016) found that the propagation time of the wind signals are much shorter than that of CSWs in the north SCS. Even though one can see isolated cases which are quite different from the dispersion relationship of CSW in Fig. 7, most of the data points are near the theoretical dispersion relation. Moreover, the stratification in the continental shelf is important for the characteristics of CSW. The sea level variation in this study shows that coastal trapped waves influenced by stratification.

## 5.2 Trapped cross-shelf structure

### 5.2.1 CSWs

As the sampling period of the altimeter satellites is about 10 d, the signals with periods shorter than 20 d could not be distinguished from the along-track SLA based on the Nyquist sampling theorem. Fortunately, the cross-shelf structure of CSWs could be sampled as fragments by repeat satellite observations.

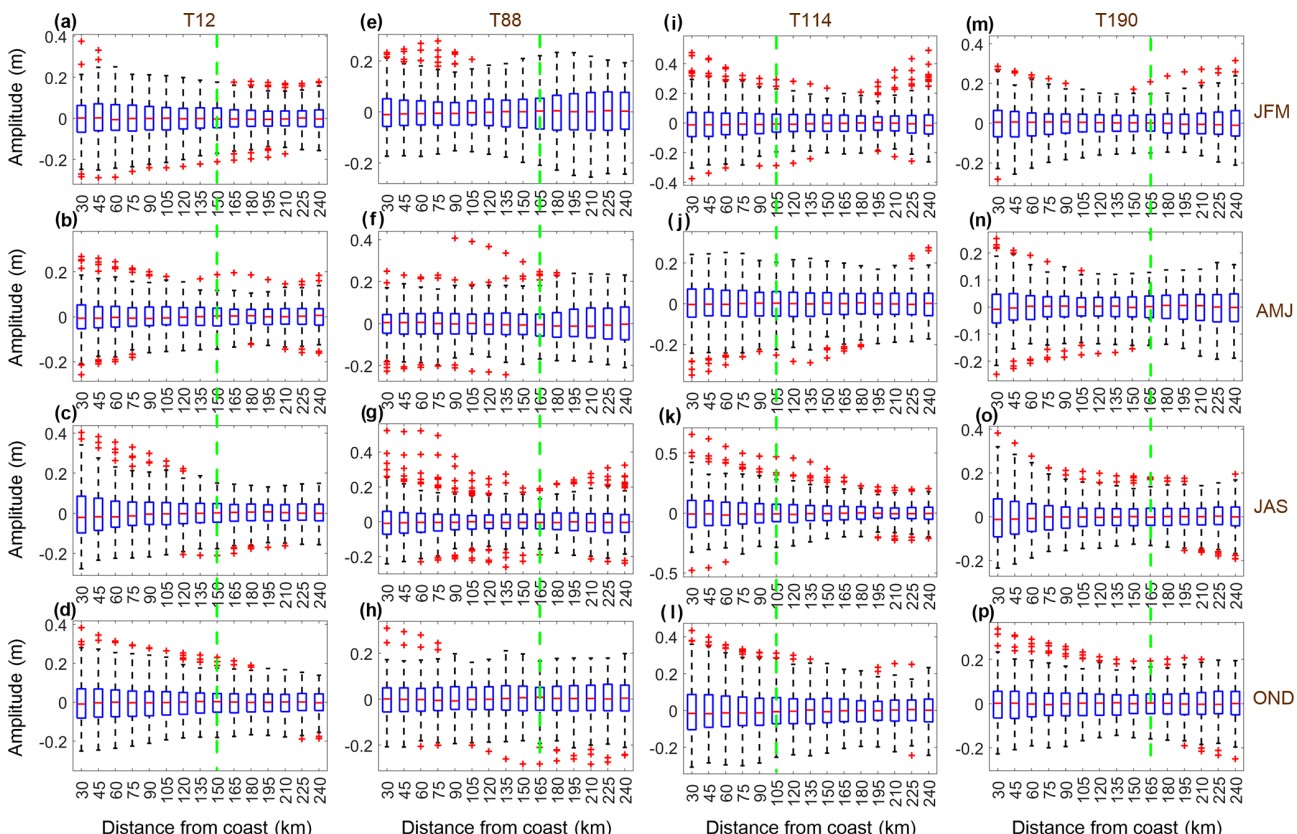

**Figure 8.** Boxplot of the along-track SLA on the shelf for tracks 12, 88, 114 and 190 from 1993 to 2020 (columns). Seasonal means of the SLA are plotted in the rows. Green dashed lines present the minimum SD of the along-track SLA. The climatological seasonal mean is removed. The along-track SLA is averaged for each 15 km offshore. In each box, the central red line indicates the median, and the bottom and top edges of the blue box indicate the 25th (Q1) and 75th (Q3) percentiles, respectively. The upper (Q3+1.5IQR) and lower (Q1-1.5IQR) whiskers extend to the most extreme data points not considered outliers. The outliers are the most extreme data points (larger than upper whisker or smaller than lower whisker), plotted individually using the red cross marker. IQR = Q3 − Q1.

In this case, the CSWs with a period of less than 40 d were abundant from 1993 to 2020 on the northern continental shelf of the SCS, especially in winter (Fig. 5). Even if the significance level is larger than 5 %, the power is universal and continuous in this period band. Therefore, it could be considered that a large number of repeat observations by altimeter satellites were executed during CSW events.

Figure 8 shows a boxplot of the along-track SLA over the continental shelf for tracks 12, 88, 114 and 190. The maxima, minima and outliers derived from Fig. 8 are listed in Table 2. The dashed black whisker line outside of the box extends to the most extreme data points. One can see that the maximum amplitude (75th percentile), interquartile range (IQR) and outliers (red cross sign) occur at the coastal side. In contrast, the minima occur at the edge of the continental shelf (150, 165, 105 and 165 km offshore for tracks 12, 88, 114 and 190). The amplitude of the along-track SLA for maxima and IQR decreases gradually from the coastline (0.2–0.4 m) to the edge of the continental shelf (∼ 0.1 m). The largest outlier is 0.65 m over the shelf 30 km offshore.

The largest SLAs over the shelf occur from July to September. For example, the 75th percentile of SLAs near the coastline is about 0.25 m for track 144 and 0.42 m for track 88. The largest outlier of 0.65 m also occurs from July to September for track 88. The smallest SLA occurs from April to June when the wind is weak during the monsoon transition period (Wang et al., 2009). The extreme data from July to September show the occurrences of storm surges over the shelf (Chen et al., 2014).

Overall, the maximum amplitude, IQR and extreme data of the along-track data over the shelf show the trapped wave characteristics: the SLA decreases gradually from the coastline to the edge of the continental shelf. The trapped characteristics from the along-track SLA are similar to the cross-shelf structure of normalized amplitudes of the Mode 1 CSWs, as shown in Fig. 3a. The along-track SLA should contain higher modes. The along-track SLAs along tracks 153 and 229 show similar characteristics (not shown). The SLA along track 240 (as shown in Fig. 6n) presents a differentiated pattern in the coast side and shelf edge during May–July. The

**Table 2.** Parameters extracted from along-track SLAs in Fig. 8.

| Position[a] | Track (distance)[b] | T12 (150 km) | | | | T88 (165 km) | | | | T114 (105 km) | | | | T190 (165 km) | | | |
|---|---|---|---|---|---|---|---|---|---|---|---|---|---|---|---|---|---|
| | Month | JFM | AMJ | JAS | OND | JFM | AMJ | JAS | OND | JFM | AMJ | JAS | OND | JFM | AMJ | JAS | OND |
| Coast | Outlier (m) | 0.28 | 0.25 | 0.38 | 0.34 | 0.47 | −0.35 | 0.65 | 0.43 | 0.23 | 0.24 | 0.52 | 0.31 | 0.38 | 0.27 | 0.41 | 0.38 |
| | Max (m) | 0.26 | 0.19 | 0.32 | 0.24 | 0.29 | 0.24 | 0.42 | 0.36 | 0.21 | 0.17 | 0.25 | 0.18 | 0.21 | 0.19 | 0.34 | 0.28 |
| | Min (m) | −0.23 | −0.22 | −0.24 | −0.23 | −0.3 | −0.24 | −0.33 | −0.31 | −0.17 | −0.18 | −0.24 | −0.2 | −0.25 | −0.21 | −0.28 | −0.25 |
| Edge | Outlier (m) | 0.17 | −0.15 | 0.19 | 0.19 | 0.23 | – | 0.29 | – | 0.21 | 0.39 | 0.28 | 0.16 | −0.2 | 0.2 | −0.2 | 0.21 |
| | Max (m) | 0.15 | 0.12 | 0.13 | 0.16 | 0.16 | 0.2 | 0.2 | 0.22 | 0.17 | 0.18 | 0.15 | −0.18 | 0.16 | 0.12 | 0.14 | 0.17 |
| | Min (m) | −0.15 | −0.14 | −0.13 | −0.15 | −0.19 | −0.2 | −0.2 | −0.21 | −0.16 | −0.14 | −0.15 | −0.2 | −0.17 | −0.13 | −0.16 | −0.17 |

[a] Coast is the position with the highest SD for each track, about 15 km offshore. Edge is the position with the minimum SD of the along-track SLA as shown in Fig. 8. [b] Distance between coastline and edge.

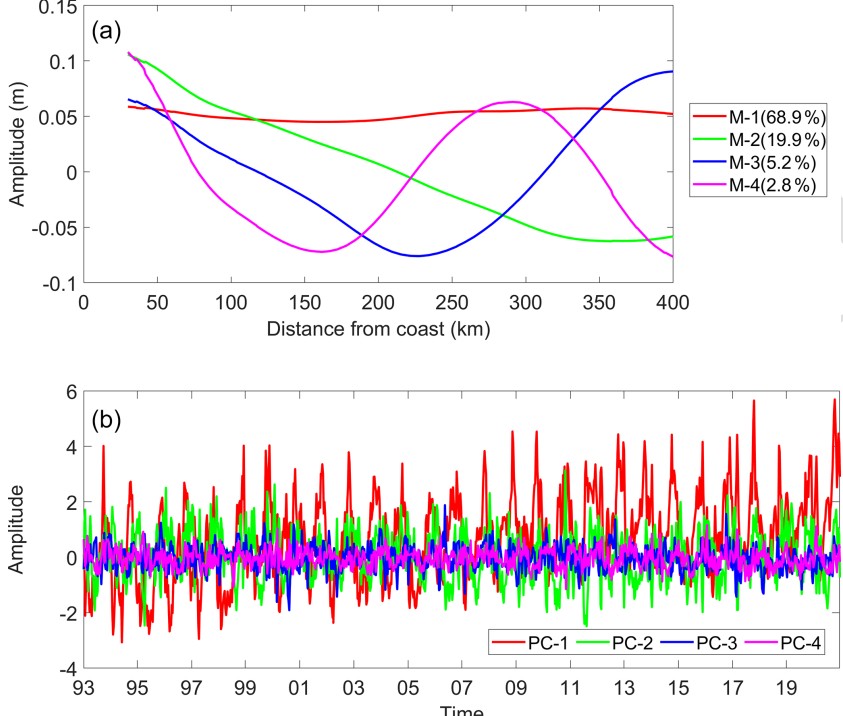

**Figure 9.** The first four EOFs for the along-track SLA on the shelf along track 12. The cross-shelf amplitude **(a)** and time series **(b)** of EOFs. The variance explained by each mode is labeled in **(a)**.

main reason is likely to be the existence of a cold eddy north of Taiwan Island.

To further reveal the variations in the along-track SLA in track 12 on the shelf, the empirical orthogonal function (EOF) analysis results are shown in Fig. 9 (EOF analysis for data along other tracks is similar; not shown here). The first four EOF modes of the along-track SLA explain 96.8 % of the total variance. Mode 1 explains the seasonal variance of SSH (red curve in Fig. 6o–u). The seasonal cycle could be characterized clearly from Fig. 9b, with peaks in October and troughs in May. Mode 2 and Mode 3 are similar to the cross-shelf structure as shown in Fig. 3a; they explain 25.1 % of the total variance. Mode 3 is influenced by the background of SSH on the open-sea side. The amplitude is 0.3–0.4 m, which is comparable to the outlier data as shown in Table 2. Mode 4 is similar to the cross-shelf structure of the Mode 2

CSW as shown in Fig. 3b; it only explains 2.8 % of the total variance. The EOF analysis indicates that the CSW could explain < 30 % of the total variance of SSH in the study area, which is comparable to the IQR in the boxplot of the along-track SLA (Fig. 8).

## 5.3 ATWs

As shown in Fig. 6, the inter-seasonal and seasonal signals could be distinguished from along-track SLAs. Figure 10 shows the seasonal mean of the along-track SLA and along-shelf sea surface wind. One can see that the time series of the seasonal mean of the along-track SLA show the seasonal variation with the amplitude of 0.1–0.2 m at 15 km offshore. The time series of the seasonal mean of the DSLA show similar characteristics with the along-track SLA. The correlation relationship between the seasonal mean of the along-

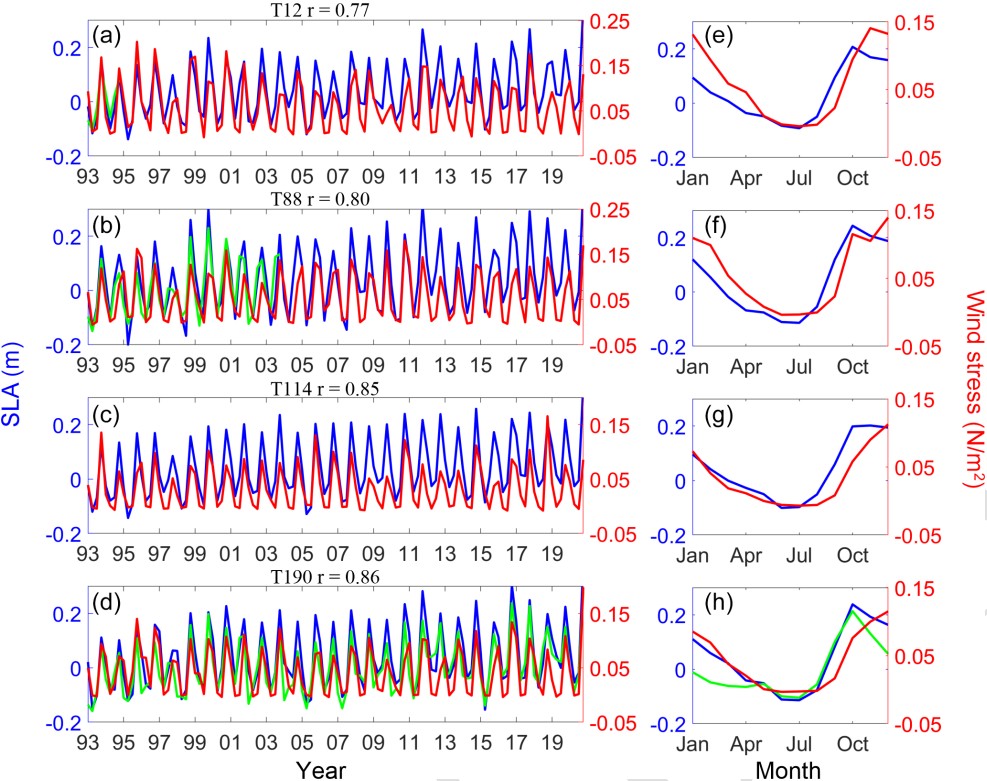

**Figure 10.** Time series of the seasonal mean of the along-track SLA (blue) and along-shelf sea surface wind stress (red) for tracks **(a)** 12, **(b)** 88, **(c)** 114 and **(d)** 190 and **(e–h)** the climatological monthly mean of the along-track SLA and along-shelf sea surface wind stress. Blue curves represent the along-track SLA at 15 km offshore. Green curves represent the seasonal mean of sea level data at tide-gauge stations **(a)** Xiamen, **(b)** Shanwei and **(d)** Zhapo. The green curve in **(h)** is the climatological monthly mean of sea level data at tide-gauge station Zhapo.

track SLA and along-shelf sea surface winds reaches >0.77. The monthly mean of the SLA during April–September is negative, while it is positive in the other months. This is attributed to that the local wind stress substantially influencing the coastal sea level (Lin et al., 2022).

In addition, one can also see an out-of-sync characteristic between the seasonal mean of the along-track SLA and wind. The maximum mean sea surface wind stress occurs in November and December, while that of monthly along-track SLA occurs in October. Ding et al. (2020) investigated the seasonality of coastal circulation in the north SCS using a numerical model. The result indicates that the maximum of water transport is 1.93 Sv, occurring in autumn. Li et al. (2023a) found the along-shelf current is strongest in October at approximately $0.17\,\mathrm{m\,s^{-1}}$. The along-shelf current involves the ATW, which is the reason why there is a difference between the monthly climatological mean of the along-track SLA and sea surface wind stress as shown in Fig. 10e–h.

Figure 11 shows the cross-shelf structure of the seasonal mean of the along-track SLA. One can see that the SLA on the coastline side is lower than on the ocean side from April to September. In the other seasons, the slope of the along-track SLA is the opposite. The along-track SLA presents sim-ilar characteristics as shown in Fig. 10; i.e., the variation in the SLA is controlled by the sea surface winds over the shelf.

Meanwhile, from Fig. 11 one can see the cross-shelf structure of the along-track SLA. The fitting curves show that theoretical ATWs (Fig. 4) explain the cross-shore structure of the along-track SLA very well. The amplitudes of ATWs in track 12 are 0.04, −0.06, −0.05 and 0.10 m in January–March, April–June, July–September and October–December. That along track 88 is relatively larger, e.g., 0.13 m in October–December, while the minimum amplitude occurs in track 114. Differently from Fig. 8, the amplitudes of ATWs during October–December are larger than that during July–September. This can be attributed to monsoon winds in winter being stronger than those in summer.

Lin et al. (2021) investigated the tilt of mean dynamic topography along the coast of the Chinese mainland. Wu (2021) used a nondimensional parameter ($\mathrm{Pe}_\beta = D_\beta/e$) to describe the influence of open-ocean forcing on shelf circulation, which is determined by the ratio of long-wave-limit planetary wave to TRW speeds ($D_\beta$) and a linear Ekman number ($e$). In this study, $\mathrm{Pe}_\beta < 1$, which indicates that shelf currents decayed rapidly toward the coast. Both the results

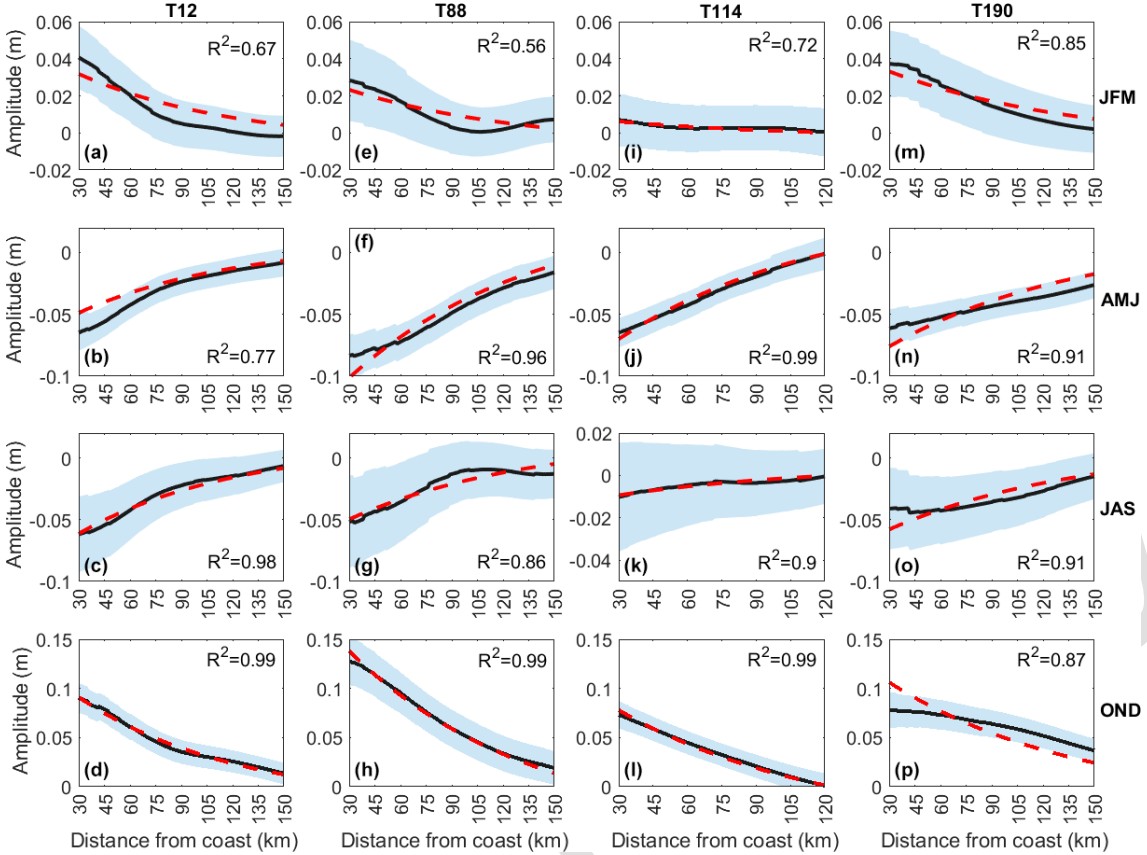

**Figure 11.** Fitting climatologic seasonal mean of the along-track SLA (black curves) using cross-shelf ATWs (red dashed curves) for tracks **(a–d)** 12, **(e–h)** 88, **(i–l)** 114 and **(m–p)** 190. Shading around the dashed line is the SD of the seasonal mean of the along-track SLA.

confirm that the ATWs predict the coastal dynamic topography over the continental shelf of the SCS well.

## 6  Summary

Using sea level data derived from the tide-gauge stations Kanmen, Xiamen, Shanwei, Hong Kong and Zhapo, this study analyzes statistical features of the CSWs and inter-seasonal and seasonal signals. Meanwhile, along-track SLA data derived from multiple satellite altimeters from 1993 to 2020 are applied to detect the cross-shelf structures of the signals. The major results are summarized as follows.

CSWs of periods shorter than $40\,\mathrm{d}$ propagate along the coast with a phase speed of about $10\,\mathrm{m\,s^{-1}}$ in the ECS and $8\,\mathrm{m\,s^{-1}}$ in the SCS. The dispersion relation indicates that the waves belong to Mode 1 CSWs.

Owing to the fact that the repeat observation period of the satellite altimeters is comparable with that of CSWs, we combine fragments of the numerous repeat observations of the along-track SLAs to reconstruct the cross-shelf structure of CSWs. The results show that the maximum amplitudes of CSWs have remarkable seasonal variability: about $0.6\,\mathrm{m}$ during July–September but only $0.2\,\mathrm{m}$ during April–June. The reconstructed cross-shelf structures of CSWs confirm the property of Mode 1 CSWs. Moreover, the energy is trapped within the partial continental shelf shallower than $200\,\mathrm{m}$.

The inter-seasonal and seasonal signals are present as ATWs on the continental shelf. The amplitudes of ATWs have remarkable seasonal variability: $\sim 0.10\,\mathrm{m}$ during October–December and twice larger than 0.04, 0.05 and $-0.06\,\mathrm{m}$ during January–March, July–September and April–June, respectively. These results reveal that the local wind stress substantially influences ATWs on the continental shelf.

The results derived from the observation data of the along-coast tide-gauge stations combined with cross-shelf tracks of satellite altimeters are interpreted well in the framework of linear wave theory. This implies that the technological approaches developed in this study are suitable for constructing the cross-shelf structures of CSWs and ATWs on the continental shelf.

However, owing to the neglect of baroclinicity, higher modes of waves are not discussed in this paper. Observations from moorings and numerical models will be used in our future studies to obtain the characteristics of baroclinic coastal trapped waves.

*Data availability.* The tide gauge data are available at https://psmsl.org/data/obtaining/stations/933.php, https://psmsl.org/data/obtaining/stations/934.php, https://psmsl.org/data/obtaining/stations/727.php, https://psmsl.org/data/obtaining/stations/1406.php (Permanent Service for Mean Sea Level, 2023). The along-track SLAs are obtained at http://ftp-access.aviso.altimetry.fr.

*Author contributions.* JYL was responsible for writing the original draft. Review and editing were conducted by QAZ. Conceptualization was handled by JYL, QAZ and LLX. TH and YX were responsible for data curation. LLX acquired funding.

*Competing interests.* The contact author has declared that none of the authors has any competing interests.

ther geographical representation in this paper. While Copernicus Publications makes every effort to include appropriate place names, the final responsibility lies with the authors.

*Special issue statement.* This article is part of the special issue "Oceanography at coastal scales: modelling, coupling, observations, and applications". It is not associated with a conference.

*Acknowledgements.* We thank Aslak Grinsted for providing software package. We also thank the anonymous reviewers for useful comments.

*Financial support.* This research was funded by the National Key Research and Development Program of China (2022YFC3104805), the National Natural Science Foundation of China (42276019, 42176184, 41976200, 41706025), the Innovation Team Plan for Universities in Guangdong Province (2019KCXTF021), the First-class Discipline Plan of Guangdong Province (080503032101, 231420003), and the Guangdong Provincial Observation and Research Station for Tropical Ocean Environment in Western Coastal Water.

*Review statement.* This paper was edited by John M. Huthnance and reviewed by three anonymous referees.

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

**Remarks from the language copy-editor**

CE1     As this affects the contents of the whole paper, we have to ask the handling editor for approval. Please give an explanation of why this needs to be changed, which will be forwarded to the editor. Thanks.

**Remarks from the typesetter**

TS1     Please confirm.