# Peer review of "Statistical analysis of dynamic behavior of continental shelf wave"

_EGUsphere, 2023_

## Referee Comment (RC1)

**General comments:**

This study employs tidal gauge and remote sensing data to investigate the statistical characteristics of continental shelf waves in the northern South China Sea. The topic should be of interest to those who focus on the coastal dynamics. I've identified some points below which might help. I'm recommending a major revision before it can be published.

**Specific comments:**

1. According to Wang and Mooers (1976) as well as Brink (1991) and Huthnance (1995), the CSW and ATW referred in this manuscript should be more specified as Kelvin wave and topographic Rossby wave, respectively.

Wang, D., and C. N. K. Mooers, 1976: Coastal-trapped waves in a continuously stratified ocean. *J. Phys. Oceanogr.*, **6**, 853-863.

Brink, K. H., 1991: Coastal-trapped waves and wind-driven currents over the continental shelf. *Annu. Rev. Fluid Mech.*, **23**, 389-412.

Huthnance, J. M., 1995: Circulation, exchange and water masses at the ocean margin: the role of physical processes at the shelf edge. *Prog. Oceanogr.*, **35**, 353-431.

2. The description of wavelet analysis in section 2.5 is not necessary because it is a method widely used in different studies.

3. By the way, it should be better to describe the theory of CSWs in section 2 rather than in section 4.

4. The EOF analysis may be a better tool to reveal the principal modes in Figs. 3 and 8.

5. The boundary conditions should be given in sections 4.1.1 and 4.1.2.

6. Fig. 4 could be omitted because it is not helpful for the analysis and there are also few descriptions about it in the manuscript.

7. Since the authors have realized that the discrepancies in Fig. 7 may be owing to the wind forcing and baroclinicity, they had better include these effects in Eq. 6 and show the relevant dispersion curve for comparison.

8. The results shown in Fig. 9 suggest that the assumption at L260-264 is not proper for the reality.

**Technical corrections:**

1. L336: it should be (24).

---

## Author Comment (AC1)

Dear Editor and Reviewers

We are very pleased to have your comments concerning our manuscript entitled "Statistical analysis of dynamic behavior of continental shelf wave motions in the northern South China Sea" (egusphere-2023-1274). Thank the editor and reviewers for taking time out of your busy schedule to review our paper and provide constructive comments on it.

We have read and dealt with all the comments carefully. The revised manuscript with all comments highlighted with blue fronts has been uploaded, and point-to-point responses to the reviewer's comments are present following.

**Response to Comments of Reviewer 1** **(Blue font in the manuscript)**

**[Comment 1]** According to Wang and Mooers (1976) as well as Brink (1991) and Huthnance (1995), the CSW and ATW referred in this manuscript should be more specified as Kelvin wave and topographic Rossby wave, respectively.

Wang, D., and C. N. K. Mooers, 1976: Coastal-trapped waves in a continuously stratified ocean. J. Phys. Oceanogr., 6, 853-863.

Brink, K. H., 1991: Coastal-trapped waves and wind-driven currents over the continental shelf. Annu. Rev. Fluid Mech., 23, 389-412.

Huthnance, J. M., 1995: Circulation, exchange and water masses at the ocean margin: the role of physical processes at the shelf edge. Prog. Oceanogr., 35, 353-431.

**Response:** We sincerely thank the reviewer for careful reading. Though all these waves are a kind of topographic wave, we would prefer using arrested topographic wave (ATW) and continental shelf wave (CSW) in this study. The reason is as follow.

(1) We think that the definition of topographic Rossby wave (TRW) covers a wider range than that of ATW. ATW is used to describe a mean wind-driven flow trapped within a coastal zone (Csanady, 1978; Wu, 2021). The mean longshore pressure gradient sustains mean flow opposing the mean wind in this linear theory ($\partial/\partial t = 0$). However, temporal derivative term ($\partial/\partial t$) in the dynamic equations for topographic Rossby wave (TRW) is balanced by other terms, i.e., $f$ term and pressure gradient term (Hughes, 2019; Quan et al., 2021). Chen et al., (2022) pointed out that the period of TRW in the East China Sea is ~30 d. Quan et al., (2021) pointed out that the period of TRW in the abyssal South China Sea is less than 60 d. Therefore, the period of TRW is a key point. If the period of TRW is too long (~300 d in this study), then, the temporal derivative term is negligible. In this situation, it is better to use "ATW".

Moreover, this comment is very important to us. We have added some sentences and reference to make it clearly.

Line 95: The mean circulation in a coastal zone of variable depth may be modeled by linear equations (Wu, 2021).

Line 538: Wu (2021) used a nondimensional parameter (Pe$_\beta$) to describe the influence of open ocean forcing on shelf circulation, which is determined by the ratio of long-wave-limit planetary to TRW speeds. In this study, Pe$_\beta$ <1, which indicates shelf currents decayed rapidly toward the coast.

(2) Firstly, Wang et al. (1976) classified the sub-inertial waves on the coastal shelf into several types as shown in Resp_Table 1. Kelvin wave should be considered as a kind of wave propagating with a flat bottom. In this study, we use a sloping bottom (Table 1 in the manuscript), not a flat one. Secondly, the phase speed of Kelvin wave should be $\sqrt{gH}$, which is faster than that of CSW

(Huthnance, 1995). The sea level variation should contain these waves together. However, the phase speed of signals extracted from tidal gauge data is about 10 m s$^{-1}$, much less than 30 m s$^{-1}$ ($H$=100 m for $\sqrt{gH}$) in this study. Base on above reasons, we think it is better to use CSW.

As you are concerned, there are several points that need to be addressed. We have changed the description of CSW in the manuscript.

Line 36: Continental shelf wave (CSW) is a type of topographic Rossby wave (TRW) trapped in the continental shelf with amplitudes ranging from several tens' centimeters to more than one meter.

Line 51: CSWs could be taken as barotropic motion in a homogeneous coastal area. While in a stratified ocean, it could be classified into coastal trapped wave. If the bottom boundary is flat, it propagates as a Kelvin wave. Overall, they are resulting from conserving potential vorticity over the shelf (Chen et al., 2022; Quan et al., 2021; Wang and Mooers, 1976).

Resp_Table 1. The classification of coastally trapped waves. (Cited from Wang et al., 1976)

[Figure]

References:

Chen, J., Zhu, X.-H., Zheng, H., Nakamura, H., Zhao, R., Wang, M., et al., 2022: Observation of topographic Rossby waves triggered by Kuroshio path meander in the East China Sea. J. Geophys. Res., 127, e2022JC018667.

Csanady, G. T., 1978: The Arrested Topographic Wave. J. Phys. Oceanogr., 8, 47–62.

Hughes, C.W., Fukumori, I., Griffies, S.M. et al., 2019: Sea level and the role of coastal trapped waves in mediating the influence of the open ocean on the coast. Surv. Geophys., 40, 1467-1492.

Huthnance, J. M., 1995: Circulation, exchange and water masses at the ocean margin: the role of physical processes at the shelf edge. Prog. Oceanogr., 35, 353-431.

Quan, Q., Z. Cai, G. Jin, and Z. Liu, 2021: Topographic Rossby waves in the abyssal South China Sea. J. Phys. Oceanogr., 51, 1795-1812.

Wang, D., and C. N. K. Mooers, 1976: Coastal-trapped waves in a continuously stratified ocean. J. Phys. Oceanogr., 6, 853-863.

Wu, H., 2021: Beta-plane arrested topographic wave as a linkage of open ocean forcing and mean shelf circulation. J. Phys. Oceanogr., 51, 879–893.

[Comment 2] The description of wavelet analysis in section 2.5 is not necessary because it is a

method widely used in different studies.

**Response:** We agree with the comment and delete section 2.5.

**[Comment 3]** By the way, it should be better to describe the theory of CSWs in section 2 rather than in section 4.

**Response:** We have restructured the sections. Section 2 describes the observed data. Section 3 presents theory of CSWs and ATWs. Section 4 presents the characteristics of the signals derived from the tide-gauge data and along-track SLA. Section 5 discusses the CSWs detected from tide-gauge data and the cross-shelf structure of sea level over the continental shelf.

**[Comment 4]** The EOF analysis may be a better tool to reveal the principal modes in Figs. 3 and 8.

**Response:** Thanks for your valuable comment. We have added the EOF analysis into the manuscript. Line 469: To further reveal the variations in the along-track SLA in track 12 on the shelf, the empirical orthogonal function (EOF) analysis results are shown in Fig. 9 (EOF analysis for data along other tracks is similar, not shown here). The first four EOF modes of the along-track SLA explain 96.8% of the total variance. Mode 1 explains the seasonal variance of SSH (red curve in Figs. 6o-u). The seasonal cycle could be characterized clearly from Fig. 9b, with peaks in October and troughs in May. Mode 2 and Mode 3 are similar to the cross-shelf structure as shown in Fig. 3a, which explains 25.1% of the total variance. Mode 3 is influenced by the background of SSH in the open sea side. The amplitude is 0.3-0.4 m, which is comparable to the outlier data as shown in Table 2. Mode 4 is similar to the cross-shelf structure of mode-2 CSW as shown in Fig. 3b, which only explains 2.8% of the total variance. The EOF analysis indicates that CSW could explain <30% of the total variance of SSH in the study area, which is comparable to the IQR in boxplot of along-track SLA (Fig. 8).

[Figure]

Fig. 9. The first four EOFs for along-track SLA on the shelf for tracks 12. Cross-shelf amplitude (a) and time series (b) of the first four EOFs. The variance explained by each mode is labeled in (a).

**[Comment 5]** The boundary conditions should be given in sections 4.1.1 and 4.1.2.

**Response:** We think this is an excellent suggestion. We have added boundary conditions to the sections 3.1.1 and 3.1.2 (restructured manuscript following comment 3).

Line 202: which is subject to the following boundary conditions: $\phi(0) = a$, and $\phi(l) = A$. $a$ and $A$ should be arbitrary. We could take $a$ as the amplitude of oscillation in sea level from the tidal gauge station.

Line 214: which is subject to the following boundary conditions: $\phi(x \rightarrow l) = A$, and $\phi(\infty) = 0$.

**[Comment 6]** Fig. 4 could be omitted because it is not helpful for the analysis and there are also few descriptions about it in the manuscript.

**Response:** We sincerely thank the reviewer for careful reading. Fig. 4 present the phase speed of CSWs. The first zero-crossing point on the right hand of each curve points out the phase speed of mode-1 CSW. The second zero-crossing points out the phase speed of mode-2 CSW. The value of these zero-point expressed as dispersion relationship curve of CSW in Fig. 7. Therefore, Fig. 4 is very important. Thank you again for your positive comments and valuable suggestion to improve the quality of our manuscript. We have added relevant contents and revised the Fig. 4.

Line 223: The solution of phase speed for CSWs is shown as Fig. 2. The zero-crossing points for each curve present the phase speed of CSWs. The first zero-crossing point on the right hand of each curve points out the phase speed of mode-1 CSW, e.g., c = 7.6 m s-1 for track 190.

[Figure]

**Fig. 4.** Solution of phase speed for CSWs from Eq. (20). The zero-crossing values represent the phase speed of CSWs for different modes. Colorful curves are amplitude of zero-order of the first kind Bessel function in different modes for the idealized depth profile of tracks 12, 88, 114, 190, and 240, respectively. Arrows point out the phase speed of mode-1 (7.6 m s⁻¹) and mode-2 (1.6 m s⁻¹) CSWs along track 190.

Line 382: Black and green curves in (c) represent the dispersion relation (from Fig. 2) for the topographic profiles along tracks 12 and 190.

**[Comment 7]** Since the authors have realized that the discrepancies in Fig. 7 may be owing to the wind forcing and baroclinicity, they had better include these effects in Eq. 6 and show the relevant dispersion curve for comparison.

**Response:** We feel great thanks for your professional review work on our article. As you are concerned, we should consider these effects in Eq. 6. It is a big challenge for future work.

(1) wind forcing. The wind curve is ignored as the uniform along-shelf wind stress in this study. If we consider the wind stress curve, Eq. 6 should be a nonhomogeneous Bessel differential equation. Lin et al., (2005) present a fractional-calculus approach to the solutions of the classical Bessel differential equation of general order (Eq. 2.7 in Lin et al., 2005). The solution is an e-exponential form which is similar with the waveform we derived from theory and observation. Moreover, the expression from Lin et al., (2005) is very cumbersome. It is worth a detailed discussion in future.

(2) baroclinicity. In the northern SCS, the water on the shelf in winter is almost uniformly mixed under a strong monsoon. The quantity of CSW events in winter is larger than that in summer as shown in Fig. 2. It means that these sub-inertial processes should be barotropic ones, i.e., CSW. In addition, Wang and Mooers (1976) presented sea level of coastal trapped wave in cross-shelf direction (Fig. 6 in Wang and Mooers, 1976). Sea level of mode-1 wave changes little in the cross-shelf direction. Moreover, it is suitable to study these baroclinic sub-inertial processes, i.e., coastal trapped wave by using current data in whole layer. In the next work, we will analyze the baroclinic sub-inertial processes by using three moorings deployed on the continental shelf of the SCS.

We will focus on these effects in the next work. We have added shortcoming of this article in

the Summary.

Line 524: However, owing to the neglective wind stress and baroclinicity, higher modes of waves are not discussed in this paper. Observations from moorings and numerical models will be used in our future studies to obtain the characteristics of baroclinic coastal trapped waves.

References:

Lin, Shy-Der, Ling, Wei-Chich, Nishimoto, Katsuyuki, Srivastava, H. M. 2005: A simple fractional-calculus approach to the solutions of the Bessel differential equation of general order and some of its applications. Computers & Mathematics with Applications, 19(9): 1487-1498.

Wang, D., and C. N. K. Mooers, 1976: Coastal-trapped waves in a continuously stratified ocean. J. Phys. Oceanogr., 6, 853-863.

**[Comment 8]** The results shown in Fig. 9 suggest that the assumption at L260-264 is not proper for the reality.

**Response:** We agree with this comment. In fact, $\tau_0$ is variable in the space, and $\tau_s^x$ is existent. However, the simplified wind stress is helpful to solve equation. If $\frac{\partial}{\partial x}\left(\frac{\tau_s^y}{f\rho}\right)$ is considered, it is hard to derive an analytical solution. The solution of a nonhomogeneous Bessel differential equation is a form of triple integral type. Numerical model is a powerful tool to resolve the actual situation.

The main difference between monthly climatological mean of along-track SLA and along-shelf sea surface wind stress in Fig. 9 is out of sync. The maximum value of monthly mean sea surface wind stress occurs in November and December. However, the maximum value of monthly along-track SLA occurs in October. The character is consistent with previous work. Ding et al., (2020) used a Finite Volume Community Ocean Model (FVCOM) to investigate the seasonality of coastal circulation in the north SCS. He found that southwestward current in fall and winter dominates the north SCS shelf. The transport is 1.93 Sv in autumn, while it is 1.73 Sv in winter. Using geostrophic current retrieval from along-track satellite altimeter data on the shelf of the north SCS, it found that the along-shelf current is strongest in October at approximately 0.17 m s$^{-1}$ (Li et al., 2023).

Therefore, we think the main reason causing the difference should be the along-shelf current. The along-shelf current involves with ATW, which causes a difference between monthly climatological mean of along-track SLA and along-shelf sea surface wind stress as shown in Fig. 9. We have added explanation into the manuscript.

Line 508: In addition, one can also see a characteristic of out of sync between seasonal mean of along-track SLA and wind. The maximum of mean sea surface wind stress occurs in November and December. While, that of monthly along-track SLA occurs in October. Ding et al. (2020) investigated the seasonality of coastal circulation in the north SCS using a numerical model. The result indicates that the maximum of water transport is 1.93 Sv occurred in autumn. Li et al. (2023a) found the along-shelf current is strongest in October at approximately 0.17 m s-1. The along-shelf current involves with ATW, which is the reason why there is a difference between monthly climatological mean of along-track SLA and sea surface wind stress as shown in Figs. 10e-h.

References:

Ding, Y., Yao, Z., Zhou, L. et al. 2020: Numerical modeling of the seasonal circulation in the coastal ocean of the Northern South China Sea. Front. Earth Sci. 14, 90–109.

Li, J., Li, M., Wang, C., Zheng, Q., Xu, Y., Zhang, T., and Xie, L. 2023: Multiple mechanisms for chlorophyll a concentration variations in coastal upwelling regions: a case study east of Hainan Island in the South China Sea, Ocean Sci., 19, 469–484.

**[Comment 9]** L336: it should be (24).

**Response:** Thank you for your reminder. The typo is revised.

---

## Author Comment (AC2)

Dear Editor and Reviewers

We are very pleased to have your comments concerning our manuscript entitled "Statistical analysis of dynamic behavior of continental shelf wave motions in the northern South China Sea" (egusphere-2023-1274). Thank the editor and reviewers for taking time out of your busy schedule to review our paper and provide constructive comments on it.

We have read and dealt with all the comments carefully. The revised manuscript with all comments highlighted with blue fronts has been uploaded, and point-to-point responses to the reviewer's comments are present following.

**Response to Comments of Reviewer 2** (**Red** font in the manuscript)

**[Comment 1]** Introduction. Instead of very broadly describing sea level variations in the SCS and their relationship with for example ENSO, I think the authors should get quickly to the core content focused in this manuscript. For instance, the authors can review relevant studies on CSWs and ATWs in the China Seas or in global shelf seas, summarize the state-of-the-art understandings of general dynamics and regional oceanography, and propose outstanding issues particularly the ones that will be addressed in the present study. Also, listing a number of studies as in the second paragraph of Introduction is not an ideal way to summarize current understandings; rather, they should be organized in a sound and logic way.

**Response:** We appreciate for Reviewers' warm suggestion. We tried our best to improve the introduction and made some changes.

Line 36:

Continental shelf wave (CSW) is a type of topographic Rossby wave (TRW) trapped in the continental shelf with amplitudes ranging from several tens' centimeters to more than one meter (Aydın and Beşiktepe, 2022; Clarke and Brink, 1985; Heaps et al., 1988; Morey et al., 2006; Mysak, 1980; Robinson, 1964; Zheng et al., 2015). CSW is a sub-inertial motion with a wavelength much greater than the depth (Li et al., 2015; Schulz et al., 2011). It propagates along the shelf with the coast on its right (left) in the northern (southern) hemisphere (Clarke, 1977). During the impact of typhoon, an excessive flooding in the coastal zone could be induced by a propagating CSW that added to the locally wind-generated surge (Dukhovskoy and Morey, 2011; Han et al., 2012). Therefore, CSW is particularly important for coastal sea-level variations.

CSW is generally generated by large-scale weather systems moving across or along the shelf (Thiebaut and Vennell, 2010). CSW events have been reported by previous investigators lasted from 2 days to 2 weeks (Chen and Su, 1987; Li et al., 2015; Li et al., 2021; Zheng et al., 2015). The phase speed of CSWs depends on the bottom topography, ranging from 5 to 20 m s-1 (Li et al., 2015; Li et al., 2016; Shen et al., 2021). CSWs could be taken as barotropic motion in a homogeneous coastal area. While in a stratified ocean, it could be classified into coastal trapped wave. If the bottom boundary is flat, it propagates as a Kelvin wave. Overall, they are resulting from conserving potential vorticity over the shelf (Chen et al., 2022; Quan et al., 2021; Wang and Mooers, 1976).

**[Comment 2]** Section 3.1 & Fig. 2. The authors identify sea level signals with periods shorter than 40 d as CSWs while those longer than 40 d as ATWs, and state that they "show remarkably different characteristics". However, the remarkable difference is not clearly discernable from Fig. 2. In particular, I do not see evident discontinuity around the period 40 d in Fig. 2b. The authors need to illustrate more clearly what specifically the "remarkably different characteristics" Moreover, instead

of showing all the resolved periods in a single panel, I suggest plotting certain period bands of interests in different panels with enlarged views of the details and adding auxiliary lines when needed to illustrate for example the content in L168-170.

**Response:** Thanks for your comment.

(1) About "remarkably different characteristics". We explain the difference in the following sentences. As shown in Line 312 "In the period band shorter than 40 d, the signals at station Xiamen lag that at station Kamen about 15 h." and Line 318 "In the period band longer than 40 d, the phase of signals between Kanmen and Xiamen is a little complicated."

One can see the arrows point bottom left in the period less than 40 d. In the period band longer than 40 d, one can see direction of arrows is in disorder. For example, the arrows in the lower period band point left in 1996 and point down in Fig. 5d. The time lag calculated from the direction of arrows is $\Delta T = Period\frac{phase}{2\pi}$. In different periods with the same phase angle, let *period*=10 d for CSW, and *period*=100 d for ATW, then the time lag for ATW is 10 times more than that of CSW. Then, the phase speed of ATW is much less than that of CSW. The disorder of arrows for ATW indicates there is no evidence for propagating of ATW.

We think we should provide a supplementary explanation.

Line 323: In the period band shorter than 40 d, one can see the arrows point bottom left uniformly. The uniform phase lag indicates the fixed time delay of the signal between two tidal gauge station. While, the direction of arrows is in disorder in the lower period band, which indicates there is no evidence for propagating.

(2) We have added auxiliary lines into Fig. 5.

[Figure]

**[Comment 3]** This is related to the previous comment. I think the governing equations and wave solutions derived in Section 4 are more clearly for CSWs than for ATWs. This could be overcome by indicating ATWs more explicitly in Section 4.2. For example, making subsection titles of Section 4.1 and 4.2 more parallel (e.g., "Governing equations and wave solutions for CSWs/ATWs), explicitly indicating Eq. (21) is the solution for ATWs and Fig. 6 is the theoretical ATW profiles (rather than "normalized SSH ..."), etc. This would also help readers to more easily understand what theoretical ATWs refer to as described in L482.

**Response:** Thanks very much for this comment and suggestion. The title of the manuscript indicates we focus on CSW. Moreover, Csanady (1978) and Lin et al. (2021) gave an excellent result for ATW. In the manuscript, we present the cross-shelf structure of ATW using satellite observation data. We have cited these references and we think the main point is enough for analysis in the manuscript. All things considered, we added minor changes.

Line 279:

The normalized SSH in the cross-shelf direction of ATW for tracks 12, 88, 114, and 190 are shown in Fig. 4. One can see that the trapped sea level in the cross-shelf direction decays quickly from 1 at the coastline to ~0.2 at the edge of the continental shelf (~200 km), and 0.1 at a distance of 300 km. The ATW amplitude decays offshore with a scale equal to the deformation radius, and L = ~100

km in the study area, which is much less than the local Rossby radius of deformation (~600 km). Under different wind stresses, the amplitude of ATW evolves similarly to that in Fig. 3a. As Track 240 is not perpendicular to the coastline, it is beyond the scope.

**[Comment 4]** L188-190. I understand that the China Seas are overwhelmed by northeasterly in winter and by southwesterly in summer. But coastal trapped waves (CTWs) should propagate equatorward (with the coast on the right side) in any season. How come seasonal signals at Xiamen would lead that at Kanmen in summer? In which form of coastal waves would sea level variability propagate poleward from Xiamen to Kanmen? Normally when wind direction is opposed to the propagation direction of CTWs (in summer for China Seas), alongshore wind would play a limited role in regulating coastal sea level variability downstream (in the sense of propagating CTWs).

**Response:** Thanks for your valuable comment. The explanations to the comments are as follow.

As shown in Fig. 5c, one can see direction of arrows is in disorder in the lower period band. For example, the arrows point upper left in the period of 120 d in 1995. While that point down left in 1997. The disorder should be caused by the wind forcing. We have divided these seasonal signals into ATW (Line 313).

CSW will propagate along the coast from north to south as shown in Fig. 5d. No evidence indicates that CSW would propagate reversely.

**[Comment 5]** Lower, not higher, percent values of significant level mean more significant. So here I think it is "larger than" instead of "less than".

**Response:** We think this is an excellent suggestion. We have corrected that in the Lines 299 and 428.

**[Comment 6]** L246-247. The "long-wave assumption" seems to assume that the cross-shelf length is much smaller than along-shelf length ($l/L<<1$). Why would it lead to du/dt=0?

**Response:** We sincerely thank the reviewer for careful reading. Let we use a simple case:

$$\frac{\partial u}{\partial t} - fv = -g\frac{\partial \eta}{\partial x} + \frac{\tau_s^x - \tau_b^x}{\rho H} \qquad (A1)$$

$$\frac{\partial v}{\partial t} + fu = -g\frac{\partial \eta}{\partial y} + \frac{\tau_s^y - \tau_b^y}{\rho H} \qquad (A2)$$

(A1-A2) can be nondimensionalized by:

$x = lx^*$

$y = Ly^*$

$u = Uu^*$

$v = Vv^*$

$U = l/T$

$V = L/T$

$t = Tt^*$

$\delta = l/L \ll 1$

In (A1), $\frac{\partial u}{\partial t} = \frac{U}{T}\frac{\partial u^*}{\partial t^*} = \frac{l}{T^2}\frac{\partial u^*}{\partial t^*} \ll f\frac{L}{T}v^*$, therefore, it is equivalent to simply putting $\frac{\partial u}{\partial t} = 0$.

**[Comment 7]** Fig. 4 is not referenced in the main text.

**Response:** We sincerely thank the reviewer for careful reading. We have added relevant content into the manuscript.

Line 233: The solution of phase speed for CSWs is shown as Fig. 2. The zero-crossing points for each curve present the phase speed of CSWs. The first zero-crossing point on the right hand of each curve points out the phase speed of mode-1 CSW, e.g., $c$ = 7.6 m s$^{-1}$ for track 190.

Line 382: Black and green curves in (c) represent the dispersion relation (from Fig. 2) for the topographic profiles along tracks 12 and 190.

**[Comment 8]** L483-484. Descriptions in these two lines only apply to Track 12. Is this correct? If yes, this needs to be illustrated more clearly in the texts.

**Response:** Thank you for your reminder. We have corrected the information.

Line 530: The amplitudes of ATWs in track 12 are 0.04 m, -0.06 m, -0.05 m, and 0.10 m in January-March, April-June, July-September, and October-December. That in track 88 is relatively larger, e.g., 0.13 m in October-December. While, the minimum amplitude occurs in track 114.

---

## Author Comment (AC3)

Dear Editor and Reviewers

We are very pleased to have your comments concerning our manuscript entitled "Statistical analysis of dynamic behavior of continental shelf wave motions in the northern South China Sea" (egusphere-2023-1274). Thank the editor and reviewers for taking time out of your busy schedule to review our paper and provide constructive comments on it.

We have read and dealt with all the comments carefully. The revised manuscript with all comments highlighted with blue fronts has been uploaded, and point-to-point responses to the reviewer's comments are present following.

**Response to Comments of Reviewer 2** (**magenta** font in the manuscript)

**[Comment 1]** Line 108-111, there are two types of along-track SLA, i.e., SLA_unfiltered and SLA_filtered. Please clarify which kind of data in this submission has been used.

**Response:** Thanks for your comment. We have described the data more clearly.

Line 119: The along-track SLA is low pass filtered using 7-point moving average.

**[Comment 2]** Line 112, it is very important to discuss the availability of along-track SLA in the coastal zones. References: Birol F et al.. 2021 The X-TRACK/ALES multi-mission processing system: new advances in altimetry towards the coast. Adv. Space Res. 67, 2398-2415 Vignudelli S, Birol F, Benveniste J, Fu LL, Picot N, Raynal M, Roinard H. 2019 Satellite altimetry measurements of sea level in the coastal zone. Surv. Geophys. 40, 1319-1349.

**Response:** We appreciate for Reviewers' warm suggestion.

Line 122: Satellite altimetry provides a unique sea level data to the coastal sea level research. A few recent studies have stressed the importance of small-scale coastal processed on coastal sea-level variance (Cazenave and Moreira, 2022; Vignudelli et al., 2019). The along-track SLA has been successfully validated and applied to the coast zone by Birol et al. (2021). These studies present the availability of along-track SLA in the coastal zones.

References:

Birol, F., Léger, F., Passaro, M., Cazenave, A., Niño, F., Calafat, F.M., Shaw, A., Legeais, J.-F., Gouzenes, Y., Schwatke, C., Benveniste, J., 2021. The X-TRACK/ALES multi-mission processing system: New advances in altimetry towards the coast. Adv. Space Res. 67, 2398-2415.

Cazenave, A., Moreira, L., 2022. Contemporary sea-level changes from global to local scales: a review. Proceedings of the Royal Society A: Mathematical, Physical and Engineering Sciences 478, 20220049.

Vignudelli, S., Birol, F., Benveniste, J., Fu, L.-L., Picot, N., Raynal, M., Roinard, H., 2019. Satellite Altimetry Measurements of Sea Level in the Coastal Zone. Surv. Geophys. 40, 1319-1349.

**[Comment 3]** Fig. 3, give cross-shelf scales consistently in km instead of degree.

**Response:** Thanks for your valuable comment. The cross-shelf scales have been added into Fig. 6o-u.

[Figure]

[Comment 4] Line 273, add Robinson (1964) shelf wave theory.

Response: Thanks for your valuable comment. The reference has been added into the manuscript.

Line 205: The solution to Eq. (8) is expressed as the sum of the first and second kinds of Bessel functions (Robinson, 1964; Schulz et al., 2011)

[Comment 5] Line 279, the simplified bathymetry in the analytical model is not very realistic in the bathymetry of the SCS, especially in the deep ocean part. Why not use more complicated bathymetry, or realistic bathymetry by using a tool from Brink and Chapman (1985).

Brink, K. H., & Chapman, D. C. (1985). Programs for computing properties of coastal-trapped waves and wind-driven motions over the continental shelf and slope. Woods Hole Oceanographic Institution.

Response: We feel great thanks for your professional review work on our article. The simplified bathymetry is enough to the analysis of CSW in this study. There are two ways for the analysis. Firstly, it is using the tool from Brink and Chapman (1985). In the previous study (Li et al., 2023), the theoretical cross-shelf fluctuation of CSW (black curve) on the shelf agrees with the blue curve calculated from the tool. Secondly, Yin et al., (2014) used a polyline to fit the bathymetry. Both the results (in this study and Yin) are similar on the shelf. In the open sea side, the fluctuation of CSW approaches zero. The main difference between ours and theirs occurs in the shelf edge. The amplitude of CSW shown in Yin et al., (2014) is a little larger (only ~10%) than ours. Therefore, we think using a simplified bathymetry is enough.

[Figure]

Resp_Fig. 1 (a) Comparison of dispersion relation derived from this study with the Kelvin mode and the lowest mode of CSW. (b) Amplitude of sea level in cross-shelf direction. Blue curve shows the amplitude of sea level306calculated from the toolbox, black curve represents that of Kelvin mode. (c) Along-shelf velocity component in cross-shelf direction. (d) Mean depth profile. Blue curve represents mean depth profile. Black curve represents the idealized depth profile. (Cited from Li et al., 2023)

References:

Brink, K. H., & Chapman, D. C. (1985). Programs for computing properties of coastal-trapped waves and wind-driven motions over the continental shelf and slope. Woods Hole Oceanographic Institution.

Li, J., Zhou, C., Li, M., Zheng, Q., Li, M., Xie, L., 2023b. A case study of continental shelf waves in the northwestern South China Sea. Acta Ocean. Sin. Accepted.

Yin, L., Qiao, F., Zheng, Q., 2014. Coastal-trapped waves in the East China Sea observed by a mooring array in winter 2006. J. Phys. Oceanogr. 44, 576-590.

**[Comment 6]** Fig. 6, add a decay scale of the Rossby radius of deformation.

**Response:** Thanks for your valuable comment. As Rossby radius of deformation changing with the latitude and bathymetry, we have added the Rossby radius of deformation into the manuscript.

Line 282: The ATW amplitude decays offshore with a scale equal to the deformation radius, and $L$ = ~100 km in the study area, which is much less than the Rossby radius of deformation (~600 km).

**[Comment 7]** Fig. 7, lacks error bars.

**Response:** Thank you for your reminder. We have revised the figure.

[Figure]

**[Comment 8]** Data in station Kanmen was not discussed in this manuscript.

**Response:** We sincerely thank the reviewer for careful reading. The phase speed of CSW between Kanmen and Xiamen has been calculated using the tidal gauge data as shown in 4.1 and Fig. 7a. We have added the discussion for track 240 near Kanmen into the manuscript.

Line 339: While in track 240, climatological monthly mean of along-track SLA on the shelf is smaller than that in track 88 especially in July.

Line 465: The along-track SLA for tracks 153 and 229 show similar characteristics (not shown). That for track 240 (as shown in Fig. 6n) presents a differentiated pattern in the coast side and shelf edge during May-July. The main reason should be the cold eddy in the north of Taiwan Island.

---

## Author Response (AR2)

Dear Editor and Reviewer

We are very pleased to have your comments concerning our manuscript entitled "Statistical analysis of dynamic behavior of continental shelf wave motions in the northern South China Sea" (egusphere-2023-1274). Thank the editor and reviewers for taking time out of your busy schedule to review our paper and provide constructive comments on it.

We have read and dealt with all the comments carefully. The revised manuscript with all comments highlighted with blue fronts has been uploaded, and point-to-point responses to the reviewer's comments are present following.

**Response to Comments of Reviewer 1 (Red font in the manuscript)**

**[Comment 1]** Introduction. The authors added some basics of continental shelf waves together with some references (the first two paragraph), which I think is fine. But I still think it is not necessary to talk about the relationship between sea level variations in the SCS with ENSO; the ENSO-related variability is not mentioned in the rest of the manuscript except the Introduction.

(Editor: I agree. If you wish to keep this in the Introduction then you should have something new about the relation with ENSO also in the Discussion. I also agree with the following referee comments).

**Response:** We feel great thanks for your professional review work on our article. It is not easy to analyze the ENSO's impact on CSWs based on the main result. Discussion for ENSO could be carried out from Figure 9-10. However, these results are the secondary outcome. After careful consideration, the references have been deleted. The others, involving seasonal variability of sea level, are kept. We hope we could analyze the relationship between CSW and ENSO in future.

Deleted sentences:

Rong et al. (2007) investigated the relationship between ENSO (El Niño and Southern Oscillation) and interannual variability of sea level in the SCS.

Wang et al. (2017) found that seasonal level anomalies are closely related to ENSO events (Wang et al., 2022).

**[Comment 2]** Fig. 5. The authors added two sentences in L323-327 of the revised manuscript, which do help me better understand the "remarkably different characteristics" between signals with periods shorter and longer than 40d. However, the authors seem to wrongly describe the left and right directions of the arrows, both in the manuscript and the response file. For example, I see from Fig. 5d that the arrow directions are primarily pointing "down-right", instead of "bottom left" as stated in L324. (There are more such cases in the response file.) Also, the arrow directions seem more uniform in the enlarged version (Fig. 5d) than in the original full time range (Fig. 5c) for the period shorter than 40d. But even in Fig. 5d, the arrows are not "uniformly" displayed (e.g., the arrows point rightward at period ~16d from Jan to Feb 1994), then how do you obtain this constant lag period of 15h (L329) for signals at Xiamen and Kanmen?

**Response:** (1) I'm awfully sorry about the problem of the directions of the arrows. We have checked and revised the error. That is "down-right", not "bottom left".

(2) The arrows are more uniform in Figure 5d than that in Figure 5c. The reason is the resolution and length of the data that we used. The relative quantity of arrows is less for larger data. That is the issue of display. It will not affect the results. Moreover, error bars (from not uniform arrows) are shown in Figure 7.

(3) time lag=phase difference/2π×period. Simply, the arrows point down-right, i.e., π/3. The period of signals is about 100 h. The time lag is about 16.7 h.

We have added sentence into the manuscript.

Line 327: (time lag=phase difference/2π×period of signal)

**[Comment 3]** Previously I suggest making the theoretical part for CSW and ATW more parallel, but the authors explain that this manuscript focuses on CSW and hence this part leans more toward CSW. I accept that.

**Response:** Line et al. (2021) has presented an excellent result for ATWs using gridded sea surface height. In this study, we use along-track SLAs to show the cross-shelf structure of ATW. We think it is better to focus on CSWs than ATWs.

**[Comment 4]** L51-52. motion -> motions, wave -> waves.

**Response:** The typo is revised.

**[Comment 5]** L97. Change to "... a counterbalance of contributions from the along-shelf wind and bottom friction and well predicted by the ATW model".

**Response:** We have added the words "bottom friction" into the manuscript.

**[Comment 6]** L271. Change to "The solution for SSH based on Eqs. (20a-c) is (Csanady, 1978)"

**Response:** We sincerely thank the reviewer for careful reading. We have revised the sentence.

L267: "The solution for SSH based on Eqs. (20a-c) gives (Csanady, 1978)"

**[Comment 7]** L324. "... point ... uniformly" -> "... generally/primarily point ..."

**Response:** Done.

**[Comment 8]** L325. uniform -> quasi-uniform.

**Response:** We agree with this comment. It is done.

**[Comment 9]** L326. While -> However

**Response:** The typo is revised.

**[Comment 10]** L326. "in the lower period band" -> "for the period band longer than 40 d"

**Response:** Done. The expression is clearer now.

**[Comment 11]** L329. Kamen -> Kanmen

**Response:** The typo is revised.

**Response to Comments of Editor (Blue font in the manuscript)**

**Comments:**
(1) Line 53. "Overall, . . ." This sentence should exclude flat bottom boundaries.
**Response:** Thanks! Kelvin wave is a kind of gravity wave. The former sentence has been deleted from the manuscript.
"If the bottom boundary is flat, it propagates as a Kelvin wave."

**Comments:**
(2) Lines 63, 101. "repeated" –> "repeat".
(3) Line 117. "repetition" –> "repeat".
(4) Line 124. "processed" –> "processes"
**Response:** Thanks for your comment.
All "repeated" have been replaced by "repeat" in Lines 62, 97, 424, 428, 599, 560.
"repetition" has been replaced by "repeat" in Line 113.
"processed" has been replaced by "processes" in Line 120.

**Comments:**
(5) Line 176. You cannot say that "$\partial u/\partial t = 0$" but you should summarize your response to review 1 Comment 6 that the scaling implies $\partial u/\partial t \ll$ fv in (1a).
**Response:** Thanks for your useful comment. Corrections have been done.
Line 171: "Under the long-wave assumption, a nondimensionalized could be applied to Eqs. (1a-b). Then, $\partial u/\partial t \ll fv$. We neglect the $\partial u/\partial t$ term, and Eqs. (1a-b) become"

**Comments:**
(6) Line 255. "evolution" –> "structure" or "profile".
(7) Line 271. "becomes" –> "give"
(8) Lines 283-284. This is confusing. I think best to omit "equal to the deformation radius, and" in line 283.
**Response:** Done.

**Comments:**
(9) Line 318. You need to say which of Kanmen and Xiamen is leading (for pointing down) or lagging (for pointing up).
**Response:** Yes, we have added additional sentence as suggested.
Line 314: "In (d), arrows point down-right (about $\pi/3$), indicating SLA in Kanmen is leading that in Xiamen."

**Comments:**
(10) Line 326. "lower" –> "longer"
(11) Line 414. ". . Even though one can see . ."
**Response:** We have corrected and added the words as suggested.

**Comments:**

(12) Lines 417-418. I think better ". . The sea level . . should show coastal trapped waves influenced by stratification." There is only one set of waves (i.e. not "baroclinic and barotropic") and your response Table 1 precludes "baroclinic . . CSWs" all the symbols, i.e. red crosses.

**Response:** We have corrected the sentence as suggested.

Line 416: "The sea level variation in this study should show coastal trapped waves influenced by stratification."

**Comments:**

(13) Line 434. "outliers at the 5 % significance level" is unclear. How exactly is an outlier defined? Figure 8 caption needs to explain all symbols, i.e. extent of boxes, dashed lines and red crosses (better here than in the main text).

**Response:** We have corrected the sentences as suggested. We have deleted "outliers at the 5 % significance level". Outliers are the most extreme data points and explained in the figure caption.

Line 446: "In each box, the central red line indicates the median, and the bottom and top edges of the blue box indicate the 25th (Q1) and 75th (Q3) percentiles, respectively. The upper (Q3+1.5IQR) and lower (Q1-1.5IQR) whiskers extend to the most extreme data points not considered outliers. The outliers are the most extreme data points (larger than upper whisker or smaller than lower whisker), plotted individually using the red cross marker. IQR= Q3-Q1."

**Comments:**

(14) Lines 446-447. Better "The along-track SLA is averaged for each 15 km offshore"?

**Response:** Many thanks. I have replaced the sentence as suggested.

**Comments:**

(15) Please format Table 2 to optimise vertical alignment. Better "Maxi" –> "Max", Mini –> "Min".

**Response:** Thanks. I have revised the words as suggested.

**Comments:**

(16) Lines 464-465. "It should contain higher modes in the along-track SLA." –> "The along-track SLA should contain higher modes."?

**Response:** Thanks. I have replaced the sentence as suggested.

**Comments:**

(17) Figure 10 caption. What is the green curve in panel h?

**Response:** We have added the caption.

Line 510: "Green curve in (h) is climatological monthly mean of sea level data at tide-gauge station Zhapo."

**Comments:**

(18) Line 538. Please define Peβ.

**Response:** We have added information for Pe$_\beta$.

Line 543: "Wu (2021) used a nondimensional parameter (Pe$_\beta$= D$_\beta$/$\alpha$) to describe the influence of open ocean forcing on shelf circulation, which is determined by the ratio of long-wave-limit

planetary to TRW speeds $(D_\beta)$ and linear Ekman number $(\alpha)$."

---

## Author Response (AR3)

**Response to Comments of Editor (Blue font in the manuscript)**

**Comments:**
(1) Line 47. Could omit "have been".
**Response:** Thanks for your comment. Done.

**Comments:**
(2) Line 51. "homogeneous coastal area" –> "unstratified coastal sea"? ["homogeneous" needs definition; "area" might be on land.]
**Response:** Thanks for your comment. "area" has been replaced by "zone". Yes, we should use "unstratified" in the manuscript.

**Comments:**
(3) Line 52. "it could be classified into" –> "the response could be classified as"?
**Response:** Done.

**Comments:**
(4) Line 167. ". . are the surface and bottom stresses . ."
**Response:** Done.

**Comments:**
(5) Line 171. "a nondimensionalized could be" –> "scaling is"?
**Response:** Done.

**Comments:**
(6) Lines 224-225. Please check. Better "where J2 is the second order Bessel function of the first kind."?
**Response:** Done.

**Comments:**
(7) Line 227 (Equation 17). I think you need to state where J0 and J2 are evaluated.
**Response:** We have added the sentence as suggested.
Line 227: "$J_0$ and $J_2$ are known and balanced by the characteristics of CSWs and topography."

**Comments:**
(8) Lines 252-253. ". . . Nodes for Mode 2 appear . . ."
**Response:** Done.

**Comments:**
(9) Line 296. "main reason" for what?
**Response:** Many thanks. The main reason for the former sentence. I have added explanation into the manuscript.
Line 227: "That is, these signals are not so significant compared with the signals with large

amplitude."

**Comments:**
(10) Line 306. "pointed out" –> "pointing out" or "showing"
**Response:** Thanks. Done.

**Comments:**
(11) Line 307. "The thick line . . ." I think this refers to 5(a) only; please say so.
**Response:** Thanks. I have revised the sentence as suggested.
Line 307: "The thick line in (a, c, d) is a 5% significance level against red noise"

**Comments:**
(12) Line 325. "propagating" –> "propagation"
**Response:** Done.

**Comments:**
(13) Line 336. "these" –> "such" unless Csanady (1978) studied this area (I don't think he did).
**Response:** Done. Thanks for correction.

**Comments:**
(14) Line 409. ". . . wind-forced and . . ."
**Response:** Thanks. Done.

**Comments:**
(15) Line 433. Omit "the trapped characteristics as"?
**Response:** Done.

**Comments:**
(16) Table 2. Please ensure that the final version does not split (Distance)** or Outlier(m) between rows.
**Response:** Thanks for reminding. I have changed font of the text.

**Comments:**
(17) Line 465. Better ", which shows that" –> ":" ?
**Response:** Thanks. Done.

**Comments:**
(18) Line 470. Better "That" –> "SLA"
**Response:** Done.

**Comments:**
(19) Line 479. ". . Fig. 3a; they explain 25.1% . ."
**Response:** Done.

**Comments:**

(20) Lines 482-483. ". . 3b; it only . ."

**Response:** Thanks. Done.

**Comments:**

(21) Figure 10 and caption. Lines 507-508 "Red curves represent the along-track SLA . ." does not match the figure with wind stress in red and SLA in blue.

**Response:** Thanks so much. We have revised as suggested.

**Comments:**

(22) Line 522. ". . the cross-shelf structure . ."

**Response:** Done.

**Comments:**

(23) Lines 543-546. You already used "$\alpha$" as a wavenumber (line 186 et seq.) so different notation is preferable. Please also define Ekman number here since you do not have viscosity or a single length scale.

**Response:** Thanks. I have changed the letter from "$\alpha$" to "$e$".

**Comments:**

(24) Line 577. "neglective" –> "neglect of". Omit "wind stress and" – you discuss wind stress!

**Response:** Thanks so much. We have revised as suggested.